# Global radiation in a rare biosphere soil diatom

Eveline Pinseel[1,2,3,9✉], Steven B. Janssens[2], Elie Verleyen[1], Pieter Vanormelingen[1,4], Tyler J. Kohler[5,6], Elisabeth M. Biersma[7,8], Koen Sabbe[1], Bart Van de Vijver[2,3] & Wim Vyverman[1✉]

Soil micro-organisms drive the global carbon and nutrient cycles that underlie essential ecosystem functions. Yet, we are only beginning to grasp the drivers of terrestrial microbial diversity and biogeography, which presents a substantial barrier to understanding community dynamics and ecosystem functioning. This is especially true for soil protists, which despite their functional significance have received comparatively less interest than their bacterial counterparts. Here, we investigate the diversification of *Pinnularia borealis*, a rare biosphere soil diatom species complex, using a global sampling of >800 strains. We document unprecedented high levels of species-diversity, reflecting a global radiation since the Eocene/Oligocene global cooling. Our analyses suggest diversification was largely driven by colonization of novel geographic areas and subsequent evolution in isolation. These results illuminate our understanding of how protist diversity, biogeographical patterns, and members of the rare biosphere are generated, and suggest allopatric speciation to be a powerful mechanism for diversification of micro-organisms.

[1] Laboratory of Protistology & Aquatic Ecology, Ghent University, Krijgslaan 281-S8, 9000 Gent, Belgium. [2] Meise Botanic Garden, Nieuwelaan 38, 1860 Meise, Belgium. [3] Ecosystem Management Research Group (ECOBE), University of Antwerp, Universiteitsplein 1, 2610 Wilrijk, Belgium. [4] Natuurpunt, Michiel Coxiestraat 11, 2800 Mechelen, Belgium. [5] Department of Ecology, Charles University, Viničná 7, 128 44 Prague 2, Czech Republic. [6] Stream Biofilm and Ecosystem Research Laboratory, École Polytechnique Fédérale Lausanne, GR B0 422, CH-1015 Lausanne, Switzerland. [7] British Antarctic Survey, High Cross, Madingley Rd, Cambridge CB3 0ET, UK. [8] Natural History Museum of Denmark, Øster Farimagsgade 5-Building 7, DK-1353 Copenhagen, Denmark. [9] Present address: Department of Biological Sciences, University of Arkansas, 850 W Dickson St, SCEN 601, Fayetteville, AR 72701-1201, USA. ✉email: eveline.pinseel@gmail.com; wim.vyverman@ugent.be

Biogeography as a discipline is tasked with determining which factors shape the distribution of species and the composition of communities across the Earth. For micro-organisms, supposedly high-dispersal rates and large population sizes have previously led to the prediction of limited global species richness and cosmopolitan geographic distributions[1]. However, an increasing body of work indicates that micro-organisms instead display distinct biogeographies[2]. Although cosmopolitan species exist[3], geographically restricted species have been found in aquatic[4] and terrestrial[5] environments. In addition, the introduction of high-throughput DNA sequencing has identified previously undocumented diversity in multiple groups[6], most of which are present at low (<0.1%) abundances[7]. Such 'rare biosphere' taxa are important contributors to ecosystem functioning, by representing a functional pool that promotes ecosystem resilience in the face of environmental change[7], and often exhibit disproportionately high levels of activity within communities[8]. Nevertheless, despite recent advances in our understanding of microbial ecology, the nature and drivers of microbial biogeographic patterns and diversification, and how they differ between different groups of micro-organisms, remain poorly resolved, especially for members of the rare biosphere[7], and taxa inhabiting terrestrial environments[9]. This is in part due to a lack of global-scale molecular phylogenetic studies of relatively young clades.

Protists exhibit tremendous morphological and ecophysiological diversity, and comprise distantly related clades that constitute the majority of lineages across the eukaryotic tree[10]. Although crucial to ecosystem functioning in aquatic[10] and terrestrial[11] environments, they are often omitted from microbiome surveys due to costs and complications arising from the necessity of having to use different primer sets. Nonetheless, recent studies of protists have revealed the existence of (pseudo)cryptic diversity in presumably cosmopolitan taxa[5,12] indicating that their global diversity is almost certainly underestimated. In particular, diatoms, a group of Stramenopile microalgae, represent one of the most species-rich and abundant protist groups[12]. Since their evolution in the Jurassic, ~200 (million years ago) (Ma)[13], diatoms have become key players in marine and freshwater food webs, and important drivers of global primary production and the silica cycle[14]. However, diatoms are also part of the soil microbiome, fulfilling important functional roles notably in the cryptobiotic crusts of desert and polar regions[15].

Here, we investigate global patterns in the diversity and biogeography of terrestrial protists inhabiting the rare biosphere by generating a macroevolutionary framework to shed light on the diversification history of *Pinnularia borealis*, a cosmopolitan diatom species complex. Species belonging to the *P. borealis* complex predominantly live in terrestrial habitats such as moist to dry soils and mosses[16]. The *P. borealis* complex thus represents one of relatively few diatom clades adapted to terrestrial environments, and has already demonstrated utility as a model organism for protist diversity and biogeography[17]. Although seldom locally abundant, members of the complex are easily recognizable at low magnifications in light microscopy, can survive prolonged periods in suboptimal conditions including sampling recipients, and although they are generally slow growers, they are easy to maintain in culture[17].

Using a global dataset of >800 *P. borealis* strains, we use automated molecular species delimitation methods to assess species boundaries, build a species-level dataset of the complex, and establish contemporary geographic distributions of the recognized species. A species-level time-calibrated molecular phylogeny is generated to investigate patterns of past diversification and reconstruct the historical biogeography of the complex. Our study reveals unprecedented high levels of species-level diversity of a terrestrial protist on a global scale. Patterns of past diversification and historical biogeography suggest that diversification is predominantly driven by colonization of novel geographic areas and subsequent evolution in isolation.

## Results

**Global sampling of the *P. borealis* species complex**. To investigate the diversity and biogeography of *P. borealis* on a global scale, we collected >1500 environmental samples worldwide (Fig. 1). Microscopy revealed the presence of *P. borealis* in 29% of the samples in the form of dead (13%) or live (16%) cells (see Fig. 2b for a micrograph of a living cell), confirming its patchy but widespread geographic distribution. Subsequently, a subset of 132 samples in which *P. borealis* cells were observed alive or with

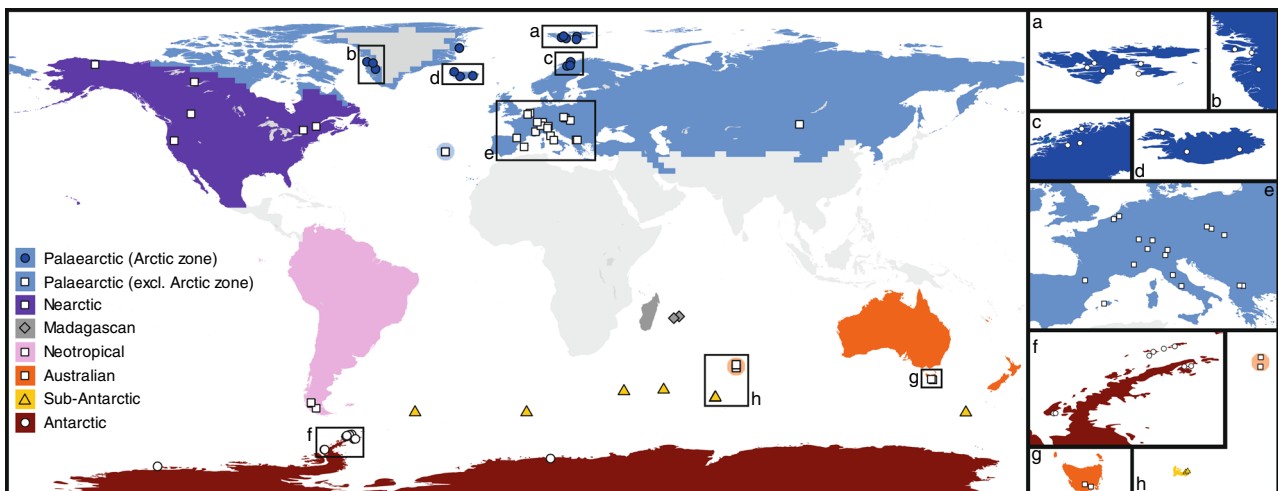

**Fig. 1 Map showing the 207 sampling locations for *P. borealis* culture material. a–h** show views of selected areas. Each individual symbol represents a sampling locality from which *P. borealis* cultures were established. Each symbol can comprise multiple individual samples. The different colors indicate eight major biogeographic regions that were based on ref. [72], in addition to the Arctic, Antarctic and Sub-Antarctic regions. These regions were used in the analysis on the historical biogeography of *P. borealis*. Samples of Amsterdam and Saint Paul Island (orange circle, **h**) were included in the same region as Tasmania (Australian zone), because they share the same climate in the Köppen-Geiger climate classification (code Cfb—temperate oceanic climate).

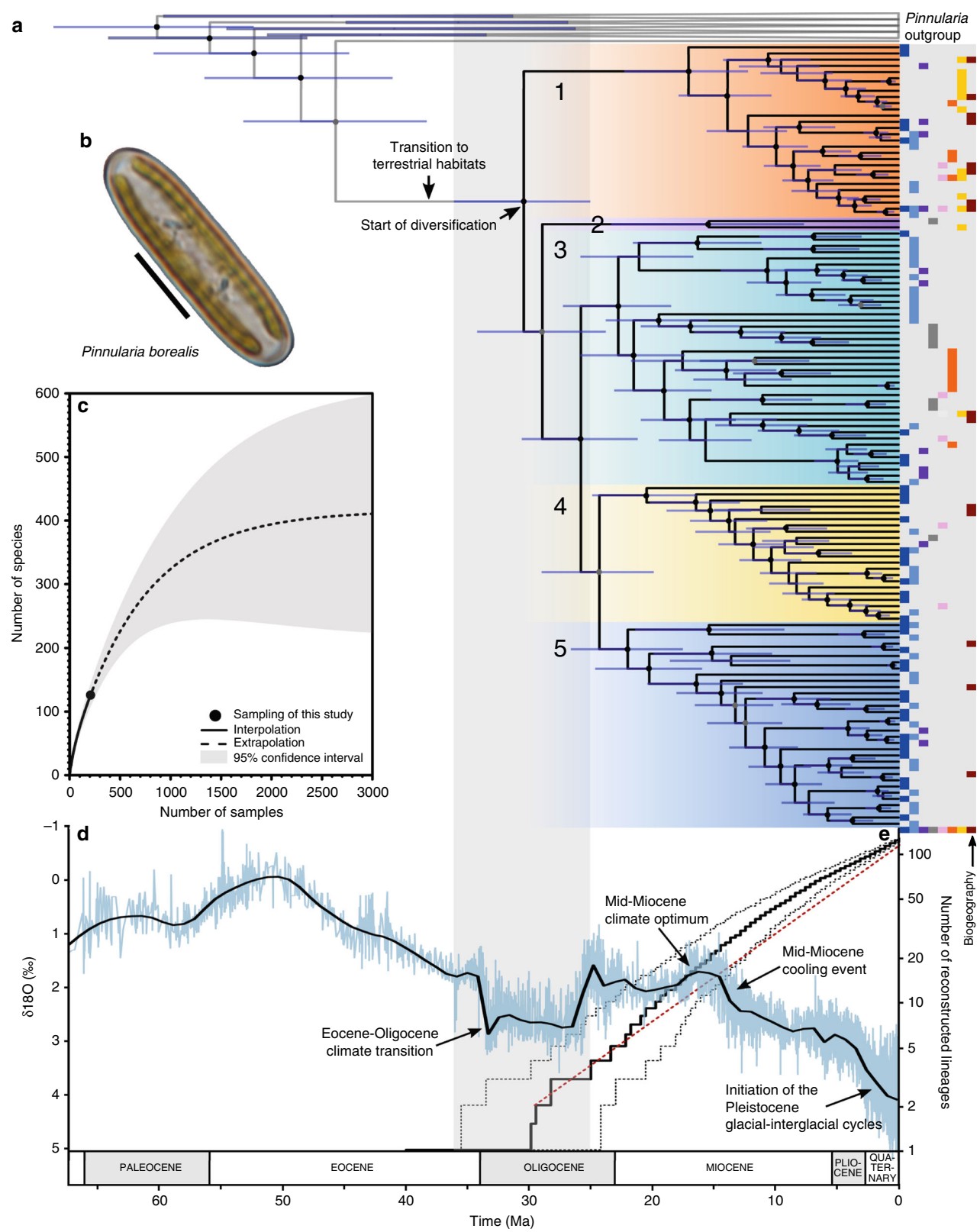

*Pinnularia borealis*

cell content, was used for environmental metabarcoding of the nuclear-encoded V4 small subunit (SSU) rDNA (18S) gene. Despite visual confirmation of the presence of *P. borealis* in these samples, *P. borealis* reads were only detected in 49 samples, and, overall, the abundance of these reads was <0.1% in more than 70% of these samples (Supplementary Fig. 1). In addition, an investigation of the global soil metabarcoding dataset obtained by ref. [9] detected no *P. borealis* reads, despite the fact that other typical soil diatoms often co-occurring with *P. borealis*, such as *Hantzschia amphioxys*[18], were retrieved. This further underscores the general rarity of *P. borealis*, and demonstrates that it is a canonical example of a rare biosphere protist.

**Fig. 2 Macroevolution of the *P. borealis* complex. a** Time-calibrated species tree of the *P. borealis* complex with indication of five major clades. The blue bars at the nodes in the phylogeny represent the 95% HPD (highest probability density) age-intervals. Coloured circles represent BEAST posterior probabilities: black ≥ 95, and grey ≥ 85. Coloured bars next to the phylogeny indicate the biogeographic region(s) of each species, using the color coding of Fig. 1 (dark blue refers to the Arctic zone). **b** Light microscopy picture of a living *P. borealis* cell in perpendicular view, showing two parallel plastids located at the cell margins. This micrograph was selected from a larger series of 178 *P. borealis* micrographs, and is representative for the general morphology of the complex. Scale bar = 10 μm. **c** Species accumulation curve showing the sample-based interpolation (rarefaction) and extrapolation of the *P. borealis* species delimited in this study. This graph shows the number of detected and expected species within the investigated regions, as well as the 95% confidence interval (grey area). Source data are provided as a Source data file. **d** Oxygen-isotope curve following the data of ref. [82], reflecting changes in global temperature and continental ice-sheet volume. The blue lines indicate the raw data, whereas the black line reflects the smoothed raw data, using a five-point running mean. **e** Lineage-through-time plot (semi-logarithmic scale) of the *P. borealis* complex (represented by the solid black line). The dashed black lines represent the 95% confidence interval, resulting from the analysis of 1000 randomly sampled post-burnin trees from the time-calibrated analysis. The dashed red line indicates the expectation under a pure-birth model, i.e. a constant rate of diversification without extinction. **a**, **e** follow the same scale: time in million years ago (Ma).

**Species-level diversity of *P. borealis*.** To unravel species-level diversity in the *P. borealis* complex, we established 867 monoclonal cultures from 207 environmental samples (Supplementary Data 1 and 2). Using Sanger sequencing, cultures were sequenced for two fast-evolving loci that are widely used for species delimitation in diatoms, including the genus *Pinnularia*: the nuclear-encoded large subunit (LSU) rDNA (28S) and the mitochondrial *cox1* genes[19]. The consensus molecular species delimitation was primarily based on *cox1*, following a conservative approach using five single-locus automated species delimitation methods. This resulted in the delimitation of 126 species (Fig. 2a, Supplementary Figs. 2–3). Species boundaries were generally congruent for 28S, although twenty-one 28S-lineages were further subdivided by *cox1* (Supplementary Fig. 3a). In several of these cases, closely related *cox1*-lineages that were lumped in the 28S-dataset showed distinct morphological differentiation, whereas more deeply diverged species were often morphologically cryptic[17] (Supplementary Fig. 3a). Such morphological divergence included occasional spine- or chain-formation, as well as large differences in cell size (Supplementary Fig. 3b). Given that these features generally coincide with species boundaries in diatoms[12], including *P. borealis*[20], all lineages delimited by *cox1* were accepted as species.

Using sample-based rarefaction analyses, we estimated the expected diversity of *P. borealis* on a global scale to equal 415 species on average (Fig. 2c). These additional species are predominantly to be expected when additional samples are investigated, as individual-based rarefaction analyses indicated that within the set of investigated samples, the overall majority of *P. borealis* species present has been found (Supplementary Fig. 4). Although our survey offers, to the best of our knowledge, the most comprehensive sampling of any diatom complex to date, and of protists in general, it is clear that extending sampling into additional regions would substantially increase the global number of known *P. borealis* species. Indeed, extrapolation of our species accumulation curves suggests that *P. borealis* strains would need to be obtained from over 1500 environmental samples before the species accumulation curve starts levelling off. Given our success rate of finding *P. borealis* in a given sample, this would imply that ca. 9500 environmental samples would need to be gathered to find the majority of *P. borealis* lineages. Clearly, species diversity in *P. borealis* largely exceeds current estimates for all other diatom taxa known to harbor (pseudo)cryptic diversity[12]. Furthermore, estimates of the expected number of species in any diatom clade with a crown age of 50 million years range up to 218[13], which is well below our average estimate for *P. borealis*.

**Patterns of past diversification.** Using the species-level dataset of *P. borealis*, we generated a multi-gene species dataset of the complex containing one representative of each species. This dataset contained two nuclear markers (28S, 18S), one

mitochondrial marker (*cox1*) and three plastid markers (*psbA*, *psbC*, *rbcL*), and was extended to include other representatives of the genus *Pinnularia*. Based on these data, we built a time-calibrated molecular phylogenetic framework to estimate the origin of diversification, to assess patterns and rates of lineage-splitting through time, and to investigate the contemporary and historical biogeography of the complex. The phylogenetic analyses generated a robust phylogeny, comprising five major well-supported clades (Fig. 2a, Supplementary Figs. 5–6). Fossil-based time-calibration of the phylogeny shows that *P. borealis* started diversifying around the Eocene/Oligocene boundary (25.0–36.1 million years ago; Fig. 2a, e, Supplementary Fig. 6). This period was characterized by the most profound change in paleoclimate throughout the Cenozoic as Earth's climate shifted from a greenhouse to an icehouse state (the EOT climate transition)[21] (Fig. 2d). This transition marks the onset of a global expansion of open landscapes[22,23], associated with colder and/or drier climates[21,24] in which *P. borealis* currently thrives.

The fossil record of *P. borealis* is rudimentary, and its uniform morphology[17] precludes direct inference of species turnover rates. Therefore, diversification rate analyses using various models were used to assess diversification rates in the complex based on the time-calibrated phylogeny. Estimating speciation and extinction rates from a phylogeny, which only contains extant diversity is a difficult task which is sensitive to undersampling, and is especially prone to underestimating extinction rates[25]. As a consequence, the inferred diversification rate estimates are subject to the shortcomings of the chosen models. To, at least partially, alleviate this issue, we chose to adopt a suite of diversification rate models, and assessed congruence amongst the results. In order to take unsampled diversity into account three sampling fractions were used, assuming complete and two scenarios of incomplete sampling. Using the approach by Magallon and Sanderson[26], we recovered net diversification rates of 0.11 (complete sampling fraction), 0.15 (30% sampling fraction) and 0.18 (10% sampling fraction) events per lineage per million year (lineage$^{-1}$ Myr$^{-1}$). These rates were very similar to those recovered by BAMM[27] (Supplementary Fig. 7a). TESS[28] recovered slightly higher net diversification rates ranging between ~0.15 and ~0.30 lineage$^{-1}$ Myr$^{-1}$ when taking unsampled species diversity into account (Supplementary Fig. 7b). The MiSSE[29] model recovered more variability compared to BAMM, with diversification rates ranging between ~0.09 and ~0.22 lineage$^{-1}$ Myr$^{-1}$ for the 30% and 10% sampling fractions (Supplementary Fig. 8a). Whereas BAMM and TESS did not detect evidence for significant rate shifts throughout the evolutionary history of *P. borealis*, MiSSE suggested that net diversification rates declined through time, and ClaDS[30] retrieved several small shifts in diversification rate throughout the phylogeny, generally resulting in a declining diversification rate through time (Supplementary Fig. 8a, b). Despite these variations

between the different models, the estimated net diversification rates were highly similar across models. In addition, the retrieved rates are in line with those observed in macro-organismal radiations[31], and well above estimates over the diatom tree as a whole, which maximally equaled ~0.06 lineage$^{-1}$ Myr$^{-1}$ [13]. However, several specific diatom clades have been found to evolve at rates that are similar to, or even exceed those of *P. borealis*[13,32]. For example, net diversification rates between ~0.075 and ~0.20 lineage$^{-1}$ Myr$^{-1}$ were reported for the eucocconeid, eunotoid and cymbelloid diatoms[13]. Previous estimates for the pinnularoid diatoms as a whole, to which *P. borealis* belongs, maximally equaled ~0.075 lineage$^{-1}$ Myr$^{-1}$ [13], suggesting that the *P. borealis* complex is evolving at a faster rate than its closest sister lineages. However, more complete taxon sampling of the entire *Pinnularia* tree as well as of other major diatom lineages is needed to gain more robust estimates of their diversification rates in order to improve the comparison with *P. borealis*.

**Transition to terrestrial habitats.** Species belonging to the diatom genus *Pinnularia* are confined to aquatic habitats, such as the shallow littoral zones of freshwater ponds and lakes, as well as semiterrestrial habitats such as wet soils and mosses[16]. In contrast, extant *P. borealis* species are found in drier, truly terrestrial, habitats[16], although some rare exceptions exist[20]. Our ancestral habitat reconstruction showed that this transition from an aquatic to a terrestrial lifestyle likely happened in the ancestor of the *P. borealis* clade (Fig. 2a, Supplementary Fig. 9). Since terrestrial environments are characterized by large diurnal and seasonal fluctuations in abiotic conditions compared with aquatic environments, such a transition must have been accompanied or preceded by ecophysiological adaptations to these more extreme conditions. Indeed, previous experiments uncovered that *P. borealis* species show high tolerance to extreme freezing (up to −180 °C)[33] and desiccation[34] events, which are lethal to other diatoms, including other *Pinnularia* species.

In this study, four *P. borealis* species were found to inhabit submerged shallow-water habitats in addition to terrestrial environments, and three were only detected in such submerged habitats, including the recently described species *P. catenaborealis*[20] (Supplementary Fig. 9). Such occurrences in aquatic environments were most pronounced in the Antarctic region, and could reflect niche availability due to reduced interspecific competition resulting from the generally low species-diversity of freshwater diatoms in this region (Verleyen and Van de Vijver, own observations). In addition, shallow ponds and the littoral zones of lakes can be highly ephemeral and susceptible to desiccation, thus representing a more terrestrial environment. Interestingly, the aquatic species *P. catenaborealis* was found to exhibit high tolerance to freezing stress[33], suggesting that although some truly aquatic *P. borealis* species might exist, they likely retained the ancestral adaptations to a terrestrial lifestyle. Indeed, our analyses suggest that occurrences of *P. borealis* in shallow aquatic environments represent recent secondary colonizations from a terrestrial ancestor (Supplementary Fig. 9). Such colonizations might have been triggered whenever the (a)biotic conditions, such as reduced interspecific competition, allowed for it.

**Contemporary and historical biogeography.** In order to account for incomplete sampling in the culture dataset, we combined the cultures and metabarcoding data (18S) to determine the geographic distributions of individual *P. borealis* species, and to assess congruence between both datasets. The metabarcoding dataset was analyzed using the R-based pipeline DADA2[35], which generates Amplicon Sequence Variants (ASVs). In contrast to

Operational Taxonomic Units (OTUs), which cluster multiple sequences together, ASVs represent unique haplotypes. This makes it possible to distinguish sequences with as little as one base pair difference. The 18S gene has limited species-level resolution in the *P. borealis* complex, as several species pairs differ by only one to three base pairs, and twelve species pairs showed identical 18S-haplotypes (Supplementary Fig. 10a, b). Therefore, only ASVs that were identical to reference 18S-sequences that were obtained by Sanger sequencing (Supplementary Data 2) were taken into account to assess distributions of individual *P. borealis* species. In case ASVs belonged to species pairs that could not be distinguished by means of the 18S-amplicon, the combined geographic distribution of both species as assessed by culture data was taken into account to assess congruence between the culture and metabarcoding dataset. In general, the geographic distributions (following the classification in Fig. 1) of individual species obtained by metabarcoding and culture data were identical, with the exception of one *P. borealis* species that was found to inhabit an additional region in the metabarcoding dataset (Supplementary Fig. 10c–e). Several 18S-haplotypes retrieved by metabarcoding differed by one to six base pairs from their most similar reference sequence (Supplementary Fig. 10e). These haplotypes could represent intraspecific or intragenomic variation in 18S. Alternatively, (part of) these unknown 18S-haplotypes could represent as yet unknown species-level diversity in the *P. borealis* complex, confirming the results of our species-accumulation curves that suggest that additional species-level diversity would be uncovered with additional sampling.

Using the culture and metabarcoding data to assess species distributions, we found the majority of examined species to be relatively restricted in their geographic distributions (Fig. 2a, Supplementary Fig. 5). However, we also recovered two species that occur in both hemispheres (Fig. 2, Supplementary Fig. 5). This is in concordance with previous results on freshwater protists that showed that closely related species can have geographically restricted as well as cosmopolitan/bipolar distributions[36]. Nevertheless, despite our unprecedented extensive sampling effort, in which more than 1500 environmental samples were investigated, large portions of the planet remain unsampled, particularly (sub-)tropical and temperate regions. The species distributions uncovered in our study thus have to be seen as approximations, and it is not unlikely that some of the species that now have seemingly restricted distributions might be geographically more widespread.

Using the geographic distributions of the individual *P. borealis* species as obtained in this study, we ran an analysis on historical biogeography in BioGeoBEARS[37], including the founder-event parameter *j*, and the distance parameter *x* (Fig. 3, Supplementary Fig. 11). These analyses uncovered that founder-event speciation plays a strong role in the *P. borealis* complex (Fig. 3, Supplementary Figs 11 and 12), indicating that anagenetic processes (range expansion/contraction) alone do not account for all changes in species distributions. Within-region speciation was at least as important as founder events, as was also observed in soil amoebae[5]. Considering the large spatial scale of our modelled geographic regions, in which some areas separated by thousands of kilometers of open ocean were merged (Fig. 1), it is not unlikely that a substantial number of such within-region speciation events were initiated in allopatry rather than in sympatry. This is in line with our finding that the three best scoring models in BioGeoBEARS contain the distance parameter *x* (Supplementary Fig. 11a). Estimates for *x* vary around 1 for all models in which it was included, indicating there is an approximate inverse relationship between dispersal probability and geographic distance (Supplementary Fig. 11a). Thus, dispersal probability of *P. borealis* decreases with

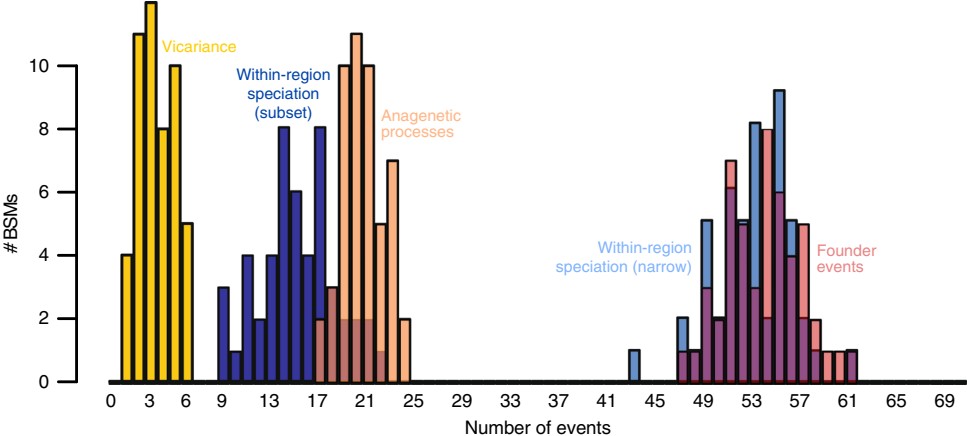

**Fig. 3 Historical biogeography of the _P. borealis_ complex.** The figure depicts the probabilities of different biogeographic processes at each node in the _P. borealis_ phylogeny, as assessed by Biogeographic Stochastic Mapping (BSM) in BioGeoBEARS. The histograms show the frequency distribution of the five different types of historical biogeographical processes included in the favored model in BioGeoBEARS (DEC $+ j + x$) in 50 BSMs. The x-axis indicates the number of events, and the y-axis indicates the number of BSMs in which a certain number of biogeographic processes was observed. A visual representation of the considered historical biogeographic processes can be found in ref. [74].

increasing geographic distance. This indicates that dispersal in the _P. borealis_ complex is limited and hence suggests that geographic isolation, and thus the conditions necessary for the (passive) divergence of isolated populations (allopatric speciation) are common. This is in line with earlier results on chrysophytes where allopatric mechanisms were suggested to play an important role in the speciation process[38]. Our analyses further suggest multiple independent colonization events of all continents by _P. borealis_. In particular, it is noteworthy that despite its geographic isolation, (Sub-)Antarctica was colonized at least eight times, and in several cases our analyses suggested this happened through long-distance dispersal from the Northern Hemisphere (Supplementary Fig. 11b).

In light of these results, it has to be noted that our extensive but still incomplete geographic sampling might have affected our results on the historical biogeographical processes shaping the evolutionary history of the _P. borealis_ complex, as anagenetic and within-region (sympatric) processes are likely to be of higher importance in a scenario with wider species distributions. Such wider species distributions may be expected to be uncovered when additional samples/regions would be investigated. However, additional sampling is also likely to uncover additional (rare) species-level diversity with restricted distributions. Furthermore, absence of unsampled species-level diversity in the _P. borealis_ phylogeny may have influenced the inference of ancestral ranges, which could in turn have impacted our estimates on historical biogeographical processes. Nevertheless, evidence for a geographic factor in the diversification of _P. borealis_ was also uncovered when investigating intraspecific diversity in one of the _P. borealis_ species. It concerns the most widely distributed _P. borealis_ species in this study, referred to by its reference strain JRI15_10_06 (clade 1). It was observed in 30% of the samples with culture material, represented 18.6% of all established cultures, and was also the most common species in the metabarcoding dataset. In addition, this species showed substantial sequence variability in the _cox1_-gene. Analysis of Molecular Variance (AMOVA)[39] revealed that this molecular variation was geographically structured between the Northern and Southern Hemisphere (Supplementary Fig. 13a, b). Molecular time-calibrated phylogenetic analysis further showed that this _P. borealis_ species comprises three distinct metapopulations (one in the Southern Hemisphere, and two in the Northern Hemisphere) that diverged during the Pleistocene, a period characterized by

repeated glacial-interglacial cycles[40] (Fig. 4, Supplementary Fig. 13c). Whereas no haplotypes are shared between the Southern and Northern Hemisphere, members of the northern metapopulations occur sympatrically in the same regions (Fig. 4, Supplementary Fig. 13b, c). These observations are concordant with geographic isolation during glacial maxima, for example in glacial refugia or nunataks, and range expansion during interglacials (a vicariance scenario), or long-distance dispersal between hemispheres followed by local divergence (a peripatric scenario). In both scenario's, the sympatric distribution of the Northern Hemisphere metapopulations resulted from secondary contact after divergence in geographic isolation. Alternatively, diversification in the Northern Hemisphere could also have occurred in sympatry.

## Discussion

Based on the results of our study, two key findings emerge: (i) _P. borealis_ shows extraordinarily high species-level diversity, most likely caused by elevated diversification rates, and (ii) diversification is predominantly driven by colonization of novel geographic areas and subsequent evolution in isolation. The existence of hyper-diverse clades is often attributed to evolutionary radiations, the most well-known example of which are adaptive radiations[41]. In general, adaptive radiations are driven by biotic factors (a key innovation), predominantly occur in sympatry, and are coupled with ecomorphological divergence[41]. There are however other types of evolutionary radiations, and in many cases multiple drivers act together in shaping diversification history[41]. Our analyses on the historical biogeography of _P. borealis_ indicate that its diversification has a strong geographical component. This indicates that allopatric speciation plays an important role, suggesting that diversification in this complex is an example of a global-scale geographic radiation. In such a radiation, a clade experiences increased opportunity for diversification due to allopatric speciation resulting from a physical barrier to gene flow[41].

Nevertheless, our results do not exclude the possibility of sympatric speciation, nor a role for divergent selection, and ultimately ecological speciation. Within-region speciation appears to be important in the _P. borealis_ complex: although it is impossible to determine the relative contributions of sympatric, parapatric and allopatric speciation within the vast continental-scale regions

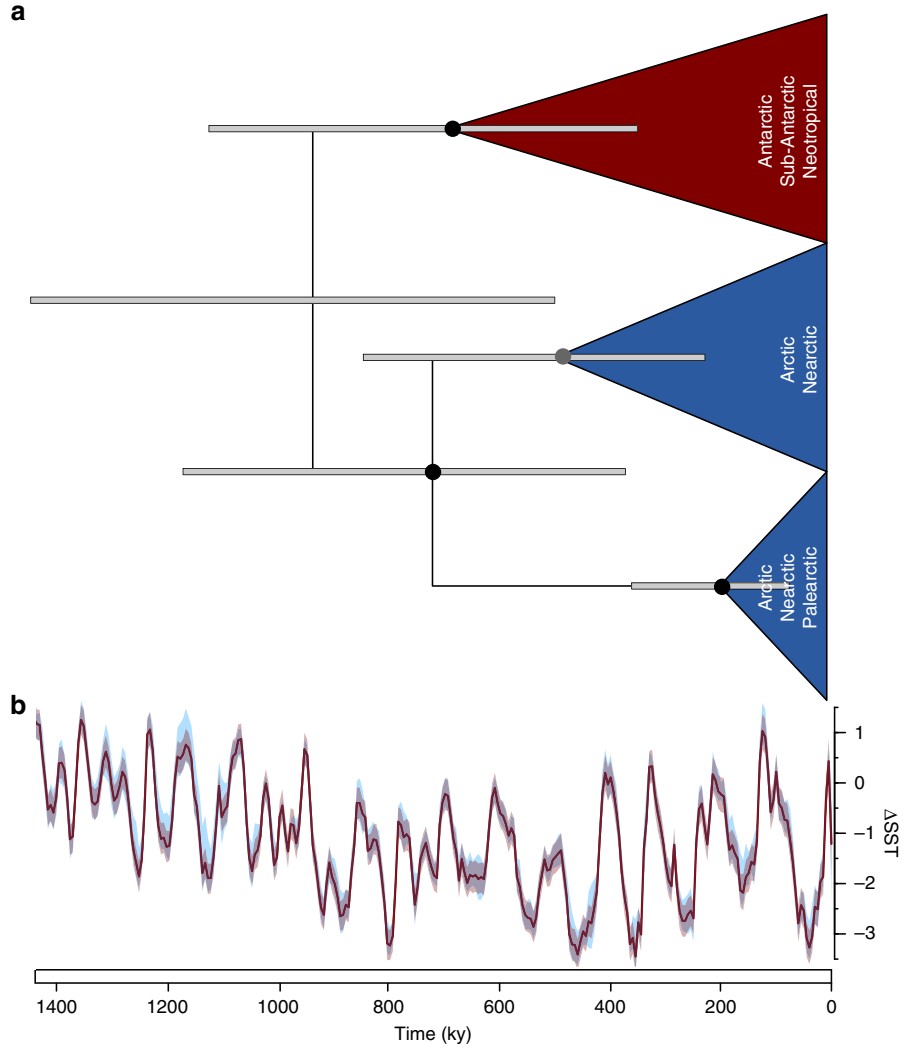

**Fig. 4 Phylogeography of a *P. borealis* species. a** Molecular time-calibrated phylogeny of *P. borealis* species JRI15_10_06, the most common and widely distributed species in this study, based on a 3-gene alignment (28S, *cox1*, *rbcL*), showing three major metapopulations. The grey bars represent the 95% HPD (highest probability density) age-intervals. Colored circles represent BEAST posterior probabilities: black > 95, and grey > 90. A more detailed phylogeny is given in Supplementary Fig. 13c. **b** Global delta sea surface temperatures (ΔSST) over the last 1.4 Ma, based on ref. [40]. The dark red line indicates the ΔSST, and the transparent bands show the 95% confidence interval based on variability (red) and jackknife (blue).

including distant islands defined in our analyses, our results at least hints towards a potential opportunity for sympatric speciation to occur. In addition, evidence for local adaptation comes from the observation that (i) different *P. borealis* species show niche-divergence regarding optimal growth temperatures and maximum temperature for growth[18], and (ii) *P. catenaborealis*, a chain-forming *P. borealis* species that is restricted to Maritime Antarctica, is likely adapted to a truly aquatic lifestyle[20]. To properly assess the role of sympatric and/or ecological speciation in the *P. borealis* complex, detailed ecological trait information should be obtained for all species, and, if possible, complemented by comparative/population genomic analyses. Nonetheless, it is highly unlikely that sympatric speciation on its own has generated the high levels of diversity of *P. borealis* observed in this study, nor that divergent selection was solely responsible for speciation after the establishment of physical barriers to gene flow. Furthermore, as noted above, it is not unlikely that different drivers are acting together in shaping the extreme species-level diversity observed in *P. borealis*.

For allopatric mechanisms to play a strong role in generating the extensive global diversity of *P. borealis*, (long-distance)

dispersal must be sufficiently frequent to allow successful colonization of new localities, while still being rare enough to prevent large-scale gene flow. In this sense, the transition to a terrestrial niche, and the associated adaptations to desiccation and freezing, could have opened up additional opportunities for allopatric speciation. On the one hand, the extreme ecophysiological tolerance of *P. borealis* undoubtedly enhances its capacity to survive (long-distance) dispersal[33,34], while on the other hand its local rarity only generates a relatively small number of colonizers over time, thus constraining gene-flow. This is in contrast with more widely distributed protists with greater population sizes, such as marine bloom-forming phytoplankton[1,3], and is in agreement with the intermediate dispersal model, which has been suggested to result in elevated diversification in macrobiota[42].

To date, we can only speculate on whether and how past environmental change might have impacted the diversification history of *P. borealis*, which started around the EOT climate transition. The EOT was characterized by a drop in global temperatures[21], and an increase in aridification[24] and seasonality[43]. It is not unlikely that these factors contributed to the dispersal and subsequent diversification of *P. borealis*. Increased aridification is

expected to increase the transport of soil particles over vast distances, possibly contributing to long-distance dispersal of soil-dwelling protists. In parallel with the decrease of forest ecosystems throughout the Miocene, grasslands, deserts and tundra vegetation expanded globally, promoted the evolution of large grazing mammals, and likely increased the availability of suitable habitats for *P. borealis*[22,23]. Today *P. borealis* is commonly found in tundra and desert ecosystems, as well as exposed environments, such as soils and moss vegetation and anthropogenically and grazer-disturbed sites. Although a direct link between past environmental change and diversification in *P. borealis* cannot be conclusively established, it is conceivable that it has contributed to the accumulation of diversity in this species complex by increasing the availability of spatial opportunity for speciation.

Evolutionary radiations of relatively young and hyper-diverse clades have long fascinated biologists[41], yet have hardly been documented for micro-organisms (but see refs. [3,44,45]). We here present a first example for a rare biosphere diatom, and propose that the multitude of terrestrial habitats on a planetary scale shapes the diversification of protists in similar ways as it does for macrobiota in geographically complex areas, and/or during periods of climatic oscillations[41]. In contrast to earlier predictions of global distributions and low species diversity in protists[1], we showed that the interplay between dispersal ability and geographic isolation has been key in generating diversity in terrestrial rare biosphere diatoms. Our dataset has provided a unique insight in the diversity of terrestrial diatom species complexes, and protists in general, and shows that when undertaking large sampling efforts, it is feasible to study rare biosphere taxa in detail.

Our work thus represents an important step forward in understanding diversity-levels and speciation in protists, and helps to obtain a more complete understanding of diversification patterns and tempo across the eukaryotic tree. Ideally, future research on protist diversity aims to include trait information in combination with diversification rate analyses, as this will be an important step towards an evolutionary and ecological approach comparing patterns and rates of diversification between closely and distantly related protist species complexes. Given the vital ecological role of terrestrial micro-organisms, understanding their diversification dynamics and geographic distributions is crucial to predict their response and safeguard biogeochemical functions in a rapidly changing world[9,11,15].

## Methods

**Field work**. In all, >1500 environmental samples were collected from terrestrial mosses, soils (top layer, ±upper 2 cm), and littoral sediments from lakes and ponds, originating from various locations worldwide. All samples were stored dark, and if possible, cool (<10 °C) during transport. From these, *P. borealis* cultures were established and used for morphological and molecular analyses. For a subset, duplicate samples were taken in the field for environmental metabarcoding. These samples were stored dark, and if possible, frozen (−20 °C). When samples could not be frozen, Sucrose Lysis Buffer (SLB; 20 mM EDTA, 200 mM NaCl, 0.75 M sucrose, and 50 mM Tris-HCl at pH 9) was added to prevent biological activity, and to preserve DNA quality. Upon arrival in the lab, all samples were frozen at −80 °C, prior to DNA extraction.

**Culture establishment**. Upon arrival in the laboratory, small quantities of the natural material (subsamples) were incubated for several weeks to months in WC medium, without pH adjustment or vitamin addition, at 4 °C (for polar and temperate regions) or 18 °C (for subtropical regions), 5–10 μmol photons m$^{-2}$ s$^{-1}$ and a 12:12h (light:dark) cycle. Although the abundances of *P. borealis* cells were low in the overall majority of the samples, this was accommodated by careful sample treatment. All environmental samples were subsampled in multiple wells of 12-well plates. In doing so, care was taken to take material from different parts of the sample, and if the sample was heterogeneous (for example, a mix of soil and moss), multiple subsamples from these different parts were taken. These samples were subsequently screened repeatedly in a light microscope over a course of several weeks. In case only dead valves of *P. borealis* were observed, samples were screened over longer time periods (up to four to six months). Although time-

consuming, this approach ensured that the chances of observing living *P. borealis* cells were maximized. Isolations of *P. borealis* cells were performed whenever it became possible to find living cells. Monoclonal cultures were established by isolating single cells under an Olympus SZX9 stereomicroscope using a needle and a micropipette. Cultures were grown in WC medium at standard culture conditions of 18 °C, 5–10 μmol photons m$^{-2}$ s$^{-1}$ and a 12:12h (light:dark) cycle, and reinoculated when reaching late exponential phase. When sufficient biomass was obtained, subsamples for morphological and molecular analysis were taken. In total, cultures were established from 207 samples (Supplementary Data 1). Previously published strains of *P. borealis*[17,18,20] were also included in the dataset, resulting in a total of 867 cultures (Supplementary Data 2).

**DNA extraction and sequencing**. Two datasets were built: a haplotype dataset (for species delimitation), and a species dataset (for phylogenetic inference). For the haplotype dataset, strains were sequenced for the D1–D3 region of the nuclear-encoded large subunit (LSU) rDNA (28S), and/or the mitochondrial *cox1* (Supplementary Data 2), as they have proven to be highly suitable for single-locus species delimitation in the genus *Pinnularia*[19]. Based on the results of the species delimitation on the haplotype dataset, a selection of strains was additionally sequenced for a (subset) of four additional genes to generate the species dataset: the V4–V9 region of the nuclear-encoded small subunit (SSU) rDNA (18S), and the plastid genes *psbA*, *psbC* and *rbcL* (Supplementary Data 2). These genes are insufficiently variable for species delimitation in *Pinnularia*[19], but sufficiently variable for phylogenetic inference in diatoms[13].

DNA extraction was performed following a bead-beating method with phenol extraction and ethanol precipitation[46]. PCR reaction mixtures for 18S, 28S (primers PBLSU1F, 1R, 2F, 2R), *cox1*, *psbA* and *psbC* contained per sample 1 μL of the template DNA, 0.5 μM of each primer, 200 μM of each deoxynucleoside triphosphate (dNTP), 0.4 μg μL$^{-1}$ of bovine serum albumin (BSA), 2.5 μL of 10x PCR buffer (Tris-HCl, (NH4)2SO4, KCl, 15 mM MgCl2, pH 8.7 at 20 °C; 'Buffer I', Applied Biosystems, Foster City, California USA) and 1.25U of Taq polymerase (AmpliTaq, Perkin-Elmer, Wellesley, Massachusetts USA). The mixtures were adjusted to a final volume of 25 μL with high performance liquid chromatography (HPLC) water (Sigma, St. Louis, Missouri USA). For 28S (primers DIR-f and T24U) and *rbcL*, 0.4 μM of each primer was used. The PCR-primer, protocols and references are listed in Supplementary Table 1a, b. In case standard PCRs failed, nested PCRs were used to obtain 28S- and *cox1*-sequences, using the same PCR settings as outlined above. Nested PCRs for 28S involved the external primers T24U and DIR-f, combined with the internal primer sets PBLSU1F/1R or PBLSU2F/2R, respectively, and PBORcox1F/1R (external) combined with PBORcox2F/2R (internal) for *cox1*. The PCR products were sequenced with their respective PCR primers and additional sequencing primers (Supplementary Table 1a). PCR products were sent for sequencing to Macrogen (http://www.macrogen.com). The obtained chromatograms were individually edited using BioNumerics v3.5 (Applied Maths, Kortrijk, Belgium).

**Preparation of the datasets**. Sequences were aligned using CLUSTAL-W as implemented in BioEdit v7.2.5[47], and subsequently manually curated. All protein coding gene sequences aligned unambiguously without any gaps. Prior to all downstream analyses, the best-fit models for nucleotide substitution were calculated (Supplementary Table 3). For the single-gene alignment of 28S the Bayesian Information Criterion (BIC) in jModelTest v2.1.3[48] was used. For the single-gene alignment of *cox1* as well all multi-gene alignments, PartitionFinder v1.1.0[49] under BIC and a greedy search algorithm was used to simultaneously assess the best-fit model for nucleotide substitution and the appropriate partition scheme. Partition testing was used to account for heterogeneity in substitution rates between and within genes, i.e. codon positions. For the six-marker dataset, a set of fourteen a priori defined partition schemes were given as input, i.e. 18S, 28S and full codon partition for all coding genes. All downstream analyses in this study were run locally, on the CIPRES Science Gateway[50], or on the IQ-TREE webserver[51].

**Automated molecular species delimitation**. Five approaches to species delimitation were applied to the two single-gene datasets of 28S and *cox1*: Statistical Parsimony Network Analysis (SPNA)[52], Automated Barcode Gap Discovery (ABGD)[53], the single threshold Generalized Mixed Yule Coalescent approach (sGMYC)[54], and the Poisson Tree Processes using both the maximum likelihood (PTP) and bayesian (bPTP) implementation[55]. These methods incorporate both distance-based approaches (SPNA, ABGD) as well as tree-based applications (sGMYC and (b)PTP). For all analyses, the full alignments of 28S and *cox1* were reduced to their unique haplotypes, resulting in 279 and 244 sequences, respectively.

SPNA was performed in TCS1.21[56] using a 95% threshold as connection limit and with gaps treated as missing data. ABGD was performed on the online webserver (http://wwwabi.snv.jussieu.fr/public/abgd/). For both 28S and *cox1*, default settings were used, except for the distance model and the number of steps, which were adjusted to K80 and 100, respectively. The X-value was adjusted depending on the dataset (1 for 28S, 1.5 for *cox1*). ABGD generates a series of species hypotheses (partitions). For the *cox1*-dataset, the initial partition was chosen, because it retrieved almost identical results as all recursive partitions,

which seemed to oversplit slightly. For the 28S-dataset, ABGD had difficulty finding a stable partition. Since the initial partition was overly conservative compared to the other automated molecular species delimitation methods, the first recursive partition was preferred, as this was the most stable partition over different prior intraspecific divergence estimates. For the sGMYC analyses, ultrametric single-gene trees were obtained in MrBayes[57], using a strict molecular clock model, a GTR + I + G substitution model and full codon partitioning. Two runs of four (three heated and one cold) Metropolis-coupled Monte-Carlo Markov Chains (MCMC) were completed for 10 million generations and sampled every 1000th generation. Convergence and stationarity of the log-likelihood and parameter values were assessed using Tracer v1.6[58]. sGMYC was performed on the 28S and cox1 consensus trees using the R-package SPLITS[59]. For the (b)PTP analysis, a maximum likelihood (ML) phylogeny was reconstructed for both the 28S and cox1 haplotype datasets using RAxML v8.2.4[60] under the same substitution models and partition schemes as for the phylogenies of the sGMYC analyses, and using 10 independent runs and 1000 pseudoreplicates. The resulting consensus tree was used as input for the (b)PTP analyses on the PTP webserver (https://species.h-its.org/ptp/). The (b)PTP analyses were run for 500,000 generations using a burnin of 25%. The trace plots were visually checked for convergence. For both genes, outgroup sequences were included in the phylogenetic analysis to allow for a correct root, but outgroup sequences were removed from the (b)PTP analyses to avoid bias due to inclusion of highly divergent sequences (following ref. [55]). Adopting a conservative approach, lineages were only accepted when delineated by at least four out of five methods, for each gene separately. The results of both genes were then compared to obtain a consensus species delimitation.

As the single gene-trees of the 28S- and cox1-haplotype datasets did not show hard conflicts, both datasets were concatenated, containing all unique sequence combinations of both genes. This resulted in a total of 347 strains (28S: 345, cox1: 325). Maximum likelihood (ML) trees were obtained in IQ-TREE v1.6.7[61]. Hundred ML optimizations and 1000 UltraFast bootstrap approximations were run using the edge-proportional model (-spp option), applying the appropriate substitution models and partition schemes as determined by PartitionFinder. All other options were left as default. Bayesian analysis (BI) was performed in BEAST v2.5.0[62] using a relaxed lognormal clock model, a Yule tree prior, and the same substitution models as the IQ-TREE analysis. Three independent runs were run for 150 million generations, sampling every 1000th generation. All runs were checked for convergence and stationarity in Tracer. Subsequently, the runs were combined, discarding 10% of the generations as burnin, all post-burnin trees were combined, and a maximum clade credibility tree with mean node heights was calculated.

**Rarefaction and extrapolation.** Individual- and sample-based interpolation (rarefaction) and extrapolation was performed in EstimateS[63]. The analyses were run separately for the results of the species delimitation of 28S and cox1. For all analyses, 100 randomizations were run and rarefaction curves were extrapolated until a total of 4000 individuals or samples was reached.

**Phylogenetic analyses.** For each species identified by the automated molecular species delimitation methods, one strain was selected for downstream analyses. ML single gene trees of all six genes (cox1, 18S, 28S, psbA, psbC, rbcL—Supplementary Table 1c) were obtained in IQ-TREE using 100 ML optimizations, 1000 standard non-parametric bootstraps (-b option) and the appropriate substitution model as estimated in jModelTest/PartitionFinder. All other options were left as default. Since the single-gene trees did not show hard conflicts, all six genes were concatenated to estimate the species tree, using the appropriate substitution models and partition schemes as determined by PartitionFinder. ML trees were obtained in IQ-TREE and RAxML. For RAxML, 10 independent runs and 1000 pseudoreplicates were used. For IQ-TREE, 100 ML optimizations under the edge-proportional model were run. IQ-TREE was run twice: once with 1000 UltraFast bootstrap approximations, and once with 1000 standard non-parametric bootstraps. All other options were left as default. BI was performed in BEAST v2.5.0 using a relaxed lognormal clock model, and a Yule tree prior. Three independent runs were run for 100 million generations, sampling every 1000th generation. All runs were checked for convergence and stationarity in Tracer with 25% of the generations discarded as burnin. All post-burnin trees were combined and a maximum clade credibility tree with mean node heights was calculated.

**Molecular time-calibration.** A time-calibrated phylogeny of the P. borealis complex was calculated in BEAST v1.10.4[64]. In order to provide a robust framework for the time-calibration, the six-marker dataset of P. borealis was integrated into the large Pinnularia tree of ref. [65] (Supplementary Table 2). Previously published Pinnularia strains[65] were resequenced to include the D3 region of 28S in the alignment, and the entire alignment was re-aligned using the methods outlined above. Four independent MCMC runs were implemented for 78–150 million generations and sampled every 1000th generation, using an uncorrelated relaxed lognormal clock model, Yule tree prior, and the appropriate substitution models and partition schemes as determined by PartitionFinder. Convergence and stationarity of all runs was checked in Tracer, after which 60% of the generations were discarded as burnin. All post-burnin trees were combined and a maximum clade credibility tree with mean node heights was calculated. For the time-calibrated

Pinnularia phylogeny, four calibration points were used using uniform prior distributions to account for the fragmentary nature of the fossil record of freshwater diatoms (Supplementary Fig. 6). The calibration strategy was based on ref. [65], updated to include newly available data and fossils (Supplementary Fig. 6b, c). The stem node of P. borealis was constrained based on the recovery of P. borealis fossils from Continental Antarctica (Supplementary Fig. 6b).

**Diversification analysis.** Lineage-through-time (LTT) plots were obtained using Phytools[66] in R (function ltt95). The analysis was run on a subset of 1000 randomly sampled post-burnin trees from the BEAST time-calibrated analysis, which allowed obtaining a 95% confidence interval. Prior to the analysis, the BEAST trees were pruned to only contain specimens from the P. borealis complex.

Net diversification rates were calculated using the bd.ms function in the R-package Geiger[67], based on ref. [26]. This function requires an extinction rate as input. Since no information is available on extinction rates in the P. borealis complex, we used the diatom-wide relative extinction rate obtained by ref. [13] as input: 0.751 lineage$^{-1}$ Myr$^{-1}$. The bd.ms function also requires an estimate of the total number of species within the dataset. Based on the extrapolation of the rarefaction curves, the average number of sampled species was estimated on 30.36%. This is likely still an underestimating on a global scale, resulting from the lack of data from large geographic areas. We therefore also ran bd.ms with a sampling coverage of 10%. Finally, we considered a complete sampling coverage to provide an absolute baseline for P. borealis diversification.

Changes in diversification rates through time were investigated using four different models: (i) the Compound poisson process on Mass Extinction Times (CoMET) analysis[68] in TESS[28], (ii) the Bayesian Analysis of Macroevolutionary Mixtures (BAMM)[27], (iii) the Missing State Speciation and Extinction model (MiSSE)[29], and (iv) the Cladogenetic Diversification rate Shift model (ClaDS)[30]. The analyses were run on the time-calibrated molecular phylogeny (maximum clade credibility tree) of the P. borealis complex, after pruning the outgroup specimens, and using the same sampling fractions as outlined above.

The CoMET analysis was run in TESS v2.1.0[28]. Empirical hyper-priors for the analyses were based on the default settings in the TESS manual, with exception of the number of expected mass extinction events ($\lambda_M$) and the number of expected rate changes ($\lambda_B$), which were both set to one. Although CoMET analyses are relatively robust to the choices of the hyper-priors, the analysis with a 30% sampling fraction was rerun using an $\lambda_B$ of two to test for robustness of the results, revealing no difference in the outcome of the analysis. The analyses were run for a maximum of 10 million reversible-jump MCMC iterations, and assuming a uniform sampling strategy.

For the BAMM v2.5.0[27] analysis, the expected number of shifts equaled one, and the starting values for the priors were estimated using the setBAMMpriors function in BAMMtools v2.1.6[27]. Four independent MCMC chains were run for 10 million generations, sampling every 1000th generation. The analyses were checked for convergence (ESS > 200), discarding 10% as burnin, and were visualized using BAMMtools.

The MiSSE model was run using the R-package hisse[29]. For each sampling fraction, four analyses were run, using one to four hidden states for the turnover fraction. The extinction fraction was not varied. All other settings were kept default. The AIC (Akaike Information Criterion) values of these models differed only with a couple of units. Therefore, the results of the four models were averaged per sampling fraction to obtain a more robust estimation of diversification rates.

At last, ClaDS is able to detect small rate shifts across a phylogenetic tree[30]. ClaDS allows to run several models, assuming extinction rates that are negligible (ClaDS0), homogeneous across all lineages (ClaDS1), or varying across lineages, but with a constant turnover (ClaDS2). We initially ran ClaDS2 using the R-package RPANDA[69] for all three sampling fractions for 500,000 generations using three chains (~3 weeks run time per analysis), after which all analyses were still far from converged. These computational limitations and convergence problems when extinction is taken into account and/or missing taxa are present in the tree are well-described in the ClaDS paper[30]. Therefore, we limited the analysis to a ClaDS0 run assuming a 100% sampling fraction. The results of ClaDS thus have to be seen as an illustration of the prevalence of small rate shifts throughout the history of P. borealis, rather than absolute values of P. borealis diversification. We ran ClaDS0 initially for 2,000,000 iterations, thinning every 200,000 iterations and using three chains. Subsequently, the gelman statistic was calculated and additional rounds of 2,000,000 iterations were added until the gelman factor was below 1.05. Following visual inspection of the MCMC chains, the first 900 recorded chain states were discarded, after which the posterior maximas of the rates were calculated. Finally, it is worth noting that at the time of publication of this manuscript, a new version of ClaDS had become available that allows for data augmentation, and promises to substantially speed-up analyses of large datasets with missing data under the ClaDS2 model.

**Historical biogeography and ancestral habitat reconstruction.** The historical biogeography of the P. borealis complex was investigated in BioGeoBEARS[37]. BioGeoBEARS combines a ML implementation of the three most commonly used models in historical biogeography and thus allows choosing the most optimal model for a given dataset: (i) the Dispersal-Extinction Cladogenesis Model (DEC), (ii) Dispersal-Variance Analysis (DIVALIKE), and (iii) the BayArea model (BAYAREALIKE) (see ref. [37] for details). For all three implementations,

BioGeoBEARS additionally allows the specification of founder-event speciation $(+j)$[37]. It is worth noting that critiques have been raised on the use of the founder-event parameter $j$ in DEC models[70], although others have argued that the latter study shows several fundamental flaws[71].

As input, BioGeoBEARS requires a time-calibrated molecular phylogeny, a set of geographical areas, and a maximum number of areas that a single species can occupy. However, increasing the number of geographic areas also increases the number of potential states (i.e. possible geographic ranges that can be occupied by a species), eventually to such an extent that an analysis cannot finish in a reasonable amount of time, or does not finish at all. Therefore, we limited our analysis to eight geographical areas following ref. [72] (Fig. 1), and allowed one species to occupy maximum six geographic areas. The latter number was based on the geographic distribution of the most widely distributed species in this study (reference strain JRI15_10_06). The geographical distributions of the species were based on the entire dataset of 867 specimens as well as the metabarcoding dataset. All six models in BioGeoBEARS (DEC$(+j)$, DIVALIKE$(+j)$, BAYAREALIKE$(+j)$)) were run with and without considering geographic distance, resulting in twelve runs. All models contained parameters $d$ (rate of range expansion) and $e$ (rate of range contraction). In addition, the distance model contained a distance-based dispersal model $(+x)$[73]. This model estimates dispersal probability as a function of distance, and requires a distance-matrix as input (Supplementary Fig. 9). The geographic distances in this matrix were obtained by calculating the minimal great-circle distance between the two closest sampling points from each pair of geographic areas as defined above. In order to avoid problems with ML optimizations, all distances were rescaled to values between zero and one, by dividing each distance by the largest distance in the matrix. The best-fit model was selected using AIC values, and the resulting ancestral state probabilities for each node were plotted on the time-calibrated phylogeny of *P. borealis*. Biogeographical Stochastic Mapping (BSM)[74] under the best-fit model allowed assessing the probabilities of different speciation events at each node in the phylogeny.

An ancestral habitat reconstruction was performed in Mesquite v3.61[75] by means of ML reconstruction using the Mk1 model. As input, the time-calibrated phylogeny of *P. borealis*, including the *Pinnularia* outgroup, was used. Three habitat types were taken into account: aquatic (submerged environments), terrestrial and shared.

**Phylogeography.** Distinct sequence clusters could be observed within the lineage with the highest amount of sequence data and the widest geographic distribution (reference strain JRI15_10_06, clade 1). AMOVA[39] was used to test whether this sequence variation was geographically structured between the Northern and the Southern Hemisphere. Since *cox1* showed distinctly more sequence variation than 28S, only the former was used for the AMOVA test. Prior to the analysis, the alignment was reduced in order to include a maximum number of sequences and variable positions, resulting in 68 *cox1* sequences belonging to 17 haplotypes. Following this reduction, all strains from some regions were removed from the dataset, but a sufficient number of strains from both hemispheres were retained to rigorously test for between-hemisphere population differentiation. AMOVA was run using the *amova* function of the R package ade4[76] and using the original, as well as clone-corrected data. The clone-corrected dataset included one representative genotype per population. Significance of the AMOVA tests was assessed using a randomization test (function *randtest*) with 999 permutations. *P*-values were corrected for multiple testing using Bonferroni correction. Haplotype networks were visualized using TCS as implemented in PopART v1.7[77].

Molecular time-calibration was used to estimate the ages of the different sequence clusters. A subset of the 28S–*cox1* haplotype alignment was obtained, covering four closely related lineages (reference strains JRI15_10_06, (Sterre6)c, MAQ17_160b_03, and JRI15_18b_09) and 48 specimens. *rbcL* was added for those strains for which it was available. The dataset was calibrated in time using BEAST v1.10.4 using an uncorrelated relaxed lognormal clock model, a coalescent constant population size tree prior, and three calibration points with uniform prior distributions (Supplementary Fig. 13). The minimum – maximum boundaries of these calibration points were based on the 95% HSP intervals of the corresponding nodes in the time-calibrated phylogeny of *P. borealis* (Supplementary Fig. 6). Three independent runs of MCMC iterations were implemented for 5 million generations and sampled every 1000th generation. Convergence and stationarity of the runs was checked in Tracer, after which 10% of the generations were discarded as burnin, all post-burnin trees were combined, and a maximum clade credibility tree with mean node heights was calculated.

**Environmental metabarcoding.** A subset of 132 environmental samples in which *P. borealis* was detected by light microscopy or by a previous yet unpublished metabarcoding analysis performed in our lab (PAE, Ghent University) was selected for environmental metabarcoding (Supplementary Data 3). The environmental samples (on average 5 g wet material/sample) were purified by coagulating and removing the extracellular DNA and proteins following ref. [78], after which the DNA was extracted using the protocol by ref. [46]. The V4-region of 18S was amplified using the universal primers TAReuk454FWD1 and TAReukREV3106[79]. PCR amplifications were performed in duplicate to reduce stochastic effects. The PCR mixtures contained per sample 1 μL of the template DNA, 0.4 μM of each primer, 200 μM of each deoxynucleotide triphosphate, 2.5 μL of 10x PCR buffer,

and 0.25U of Fast Start High fidelity Taq polymerase (Roche Inc.). The final reaction volume was adjusted to 25 μL using HPLC water. The PCR conditions were as follows: 35–40 touch-down cycles (1 min at 94 °C, 1 min at 57–52 °C and 3 min at 72 °C) with an initial denaturing step of 5 min at 94 °C and a final step of 20 min at 72 °C. All final PCR products were purified with Agencourt AMPure XP beads. Quality control was performed with a Qubit (Thermo Fisher Inc.) and BioAnalyzer (Agilent Inc.), after which the duplicates were pooled. The amplicon libraries were barcoded using the NEXTERA xt DNA kit (Illumina Inc.) following to manufacturer's instructions and purified using Agencourt AMPure XP beads. The libraries were randomly assigned to two runs, and sequenced on a 300 bp paired-end Illumina MiSeq machine at Edinburgh Genomics (http://genomics.ed.ac.uk/). The raw reads were processed into Amplicon Sequence Variants (ASVs) using the R-package DADA2 v1.6.0[35], for each run separately. The last 50 bp of the reverse readers were truncated, and the maximum expected error (*maxEE*) equaled 2. Samples were not pooled for the ASV inference step. Upon ASV inference, both runs were combined into a single dataset, and chimeric sequences were detected and removed. The taxonomic classification of the ASVs was done in Mothur[80], using a bootstrap value of 80 and the PR2 database v4.8.0[81], updated to include all available unique 18S-haplotypes of *P. borealis*, as a reference. The identified ASVs were subsequently aligned to the reference sequence of their Mothur-ID to confirm the identifications. In addition, we reclassified the ASVs obtained in the global soil metabarcoding study of ref. [9] using our updated PR2 database to search for presence of *P. borealis*.

**Reporting summary.** Further information on research design is available in the Nature Research Reporting Summary linked to this article.

## Data availability
Newly determined Sanger sequences have been deposited in GenBank under accession numbers MN319619–MN319641, MN319643–MN319644, MN319651–MN319652, MN662533, MN940449–MN940569, MN940581–MN941434, MN941851–MN941898, MN943234–MN943270, MN974675–MN974732, MN974734–MN975258, MN986897 and MN992091–MN992098. All Sanger sequences, and associated environmental data, of the *P. borealis* complex are also available on BOLD as dataset DS-PIBOR. The raw Illumina 18S-reads are available from the NCBI Sequence Read Archive under bioproject number PRJNA599198. The alignments used for the phylogenetic analyses, the phylogenetic trees, the sequences of all ASVs recovered in this study, and the ASV-table are available from Mendeley Data under https://doi.org/10.17632/9tyhcrjrnr.1. The source data underlying Fig. 2c, and Supplementary Figs 1, 4, 6c, 9, 10a, b, 11, 13a–c are provided as a Source data file. Previously published Sanger sequences are available under the GenBank accession codes listed in Supplementary Table 2 and Supplementary Data 2. The previously published metabarcoding dataset used in this study is available from Figshare under https://doi.org/10.6084/m9.figshare.7845167. Other relevant data supporting the findings of the study are available in the Supplementary Information section, the Supplementary Data files, or from the corresponding authors upon request.

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

## Acknowledgements

E.P. is a research fellow of the Belgian American Education Foundation and Fulbright Belgium. This research was supported by the Fund for Scientific Research—Flanders (FWO): funding of E.P. (aspirant grants 1104315N and 1104317N; travel grants K220116N and V443816N), P.V. (postdoctoral fellow), and E.V. (travel grant V426111N). Additional funding was provided by the: Belgian Science Policy (BELSPO) (projects MICROBIAN and CCAMBIO (SD/BA/03)); Antarctic Circumnavigation Expedition (ACE), organized by the Swiss Polar Institute founded by École Poly-technique Fédérale de Lausanne; Spanish Polar Program (in collaboration with A. Quesada); IPEV (programs 136 and 1167, in collaboration with M. Lebouvier); Polar Ecology Field Course (University of South Bohemia, Czech Republic), supported by Czech Ministry of Education (MSMT), government fund of the Czech Republic, and European Social Fund (grants LM2010009, CZ.1.07/2.2.00/28.0190 and RVO67985939). E.M.B. was supported by NERC PhD studentship (NE/K50094X/1), NERC-CONICYT grant (NE/P003079/1), Carlsberg Foundation grant (CF18-0267), logistic support from Instituto Antartico Chileno (INACH), Scientific Expedition Edgeøya Spitsbergen (SEES), and Center for Permafrost (CENPERM). T.J.K. was supported by Czech Science Foundation Junior Grant GACR 15-17346Y and Charles University Research Centre program No. 204069. We are very grateful to C. Allewaert, H. Baird, L. Blommaert, B. Chattová, S. Chown, P. Convey, B. Cuypers, I. Daveloose, S. De Decker, A. Del Cortona, F. De Vleeschouwer, D. Ertz, L. Gandois, R. Hallas, P.B. Hamilton, J. Hansen, L.H. Hansen, M. Hedblom, E. Hejduková, A. Hindáková, D.A. Hodgson, I. Hogg, C. Janion-Scheepers, I. Jüttner, J. Kavan, C. Kilroy, K. Kopalová, R. Leihy, J.J. Lembrechts, Z. Levkov, M. Loonen, B. Mariën, J. Mariën, D. McKnight, L. Meire, K. Moon, A. Peeters, B. Perren, J. Pinseel, E. Pushkareva, E. Rott, A. Sakaeva, J. Smol, B. Sivarajah, C. Souffreau, S. Srna, C.G. Steigüber, M.I. Stevens, W. Stock, K.R. Stoof-Leichsenring, M. Sweetlove, H. Tanttu, J.C. Taylor, A. Torstensson, B. Van Dam, N. Van der Putten, G. van Ee, W. Van Landuyt, W. Van Nieuwenhuyze, J. Van Wichelen and A. Wullf for providing samples. P. Siver and A.P. Wolfe provided information and material on *Pinnularia* fossils, and M. Harper and W. Dickinson provided Miocene Antarctic diatom fossils. C. Souffreau provided metadata on *Pinnularia* strains. B. Tytgat and A. Hondekyn provided technical and logistical support, respectively. S. D'hondt and T. Verstraete assisted during the lab work. We thank L. Chatrou, F. Leliaert, A. Alverson, T. Nakov and Q. Bafort for helpful discussions and advice.

## Author contributions

E.P. and W.V. designed the study and wrote the paper. E.P., W.V., E.V., P.V., E.M.B., T.J.K. and B.V.d.V. performed field work. E.P. performed lab work and analyzed the data. S.B.J. assisted with the data analysis. S.B.J., E.V., P.V., E.M.B., T.J.K., K.S. and B.V.d.V. commented on the paper and contributed significantly to discussions.

## Competing interests

The authors declare no competing interests.
