## [Peer Review File · Nature Communications]

Reviewers' Comments:

Reviewer #1:

Remarks to the Author:

This excellent manuscript represents one of the few phylogeographic studies on soil protists at a global scale. It relies on a broad sampling of a complex of cryptic species of soil diatoms, the *Pinnularia borealis* species complex, and analyses diversification patterns through geological times and across continents. It shows not only that protists have limited geographic distribution areas, but also suggests possible processes involved in speciation.

The manuscript is well-written, and the methods employed are sound and appropriate; this paper will be very influential in the field of protist biogeography and microbial biogeography in general. I would therefore recommend warmly its acceptance. I have only a couple of remarks:

Line 36 (and also Abstract): In my opinion, the question whether microbes have biogeographies or not is outdated; of course they do! Nowadays, the main question would be what the patterns are and how they are generated. I would remove the part on cosmopolitanism or, at least, give it less importance.

Line 46: a detail: It's the eukaryotic tree, the tree of life comprises all three domains of life...

Line 50: I would also cite a study concerning soil protists; there are not many, but here is one:

Line 53: a typo: and, and

Singer, D., Mitchell, E.A.D., Payne, R.J., Blandenier, Q., Duckert, C., Fernandez, L.D., Fournier, B., Hernandez, C.E., Granath, G., Rydin, H., Bragazza, L., Koronatova, N.G., Goia, I., Harris, L.I., Kajukalo, K., Kosakyan, A., Lamentowicz, M., Kosykh, N.P., Vellak, K. & Lara, E. (2019) Dispersal limitations and historical factors determine the biogeography of specialized terrestrial protists. *Mol Ecol*, 28, 3089-3100.

Line 63: please precise that the metabarcoding experiment was based on v4 SSU rRNA. Is this region variable enough to distinguish *P. borealis* spp. from other close related species?

Lines 81-82: Are there any clues for claiming that tropical regions would bring a lot of diversity? Do we expect latitudinal gradient patterns of diversity?

Line 93: As this period is more or less associated with the developing of Poaceae-dominated grasslands, I am wondering if there could be a relationship between Poaceae silica phytoliths and terrestrial diatom frustules... This is just an idea...

Line 111: Sympatric speciation has also found to be most relevant in another soil protist species complex, the testate amoeba *Hyalosphenia papilio* (Singer et al., 2019). Perhaps authors could discuss this?

Figure 1: I suggest presenting a map with the classical biogeographic realms (Neotropical, Palaeotropical, etc...). Or at least a map with realistic climatic zones that stick better to reality would make sense here.

Figure 2: Same here, I would divide into biogeographic realms.

I have no problem in signing my report: Enrique Lara

Reviewer #2:

Remarks to the Author:

This paper aims at characterizing the diversification dynamic of a soil diatoms species complex and understand the process generating this diversification dynamic. To do so, the authors (1) cultivated diatom strains from 207 samples collected worldwide (1) delimited species thanks to 2 fast evolving

loci (2) estimated the expected global diversity of the group using a rarefaction analyses (3) reconstructed a time-calibrated phylogenetic tree using 6 markers and fossils (4) analyzed the diatom species complex diversification dynamics with LTT plots and phylogenetic diversification models and (5) analyzed the ancestral biogeography of the group. The authors conclude (1) that the studied diatom group is exceptionally diverse, with high diversification rates compared to other diatom groups that would be linked to a radiation of the group after the Oligocene-Eocene cooling event and (2) that the group has a patchy distribution and is characterized by frequent colonization of novel geographical areas, emphasizing the role of allopatric speciation and putting forward the geographical radiation hypothesis.

The originality of the study lies in the analysis of diversification dynamics of terrestrial protists often omitted of microorganisms surveys. The overall message of the paper - showing how rare species complex with high species diversity can undergo geographical radiation and thus speciate in allopatry mainly, is original regarding the current literature. The paper represents a significant amount of work, is clearly written, and the results are interesting. Our main concern is that the sampling technique (cultivation) and effort (207 samples worldwide) might be responsible for the conclusions drawn by the authors, and that this possibility is not sufficiently addressed. There are other major limitations of the approaches used (detailed below) that are not even mentioned by the authors. In conclusion, we think that the paper can be a nice addition to the field if the authors can be more convincing that their results are clearly supported. Some of the conclusions will probably need to be toned-down to avoid over-statement.

Major comments :

(1) It could be that the patchiness of *P. borealis* geographic distribution comes from the sampling technique and effort. Can the authors provide justification of if/why the cultivates from the samples provide a good estimate of diatom diversity in each sample? If a strain is not cultivated in a sample, how confident can we be that it is not present? Are diatoms present in really low abundances easily cultivated? Maybe the metabarcoding data could be used to test/validate the completeness of the diversity recovered in each sample.

(2) Similarly, if a strain is not observed in one of the few samples representing a large biogeographic region, how confident can we be that it is not present in that biogeographic region? The rarefaction curves tend to suggest that the sampling effort is clearly not enough to represent the diversity within each biogeographic region, even for only cultivable strains. Absences are treated as true absences in the ancestral biogeographic reconstruction, so the false absences (linked to undersampling) are likely to bias the biogeographic analyses, overestimating the number of dispersal events. The authors could for example use sub-sampling to assess the robustness of their ancestral biogeographic reconstructions to sampling effort.

(3) One of the main results put forward by the authors is that *P. borealis* shows extraordinarily high species-level diversity, and elevated diversification rates, compared to other diatom groups. This is based on comparisons of estimates of species richness and diversification rates from Ref 12. However, a recent paper by Lewitus et al. (Nature Ecol Evol 2018) reports diversification rates for diatoms as a whole and various diatom groups that are similar to the rates reported here (see their Fig 1b & 3b). Some clades even have much higher diversification rates.

(4) Some of the diversification results are based on BAMM analyses. BAMM has been heavily criticized (e.g. Moore et al. PNAS 2016) and alternatives have been developed (Barido-Sottani et al. BioRxiv 2018, Maliet et al. Nature Ecol Evol 2019, Hohna et al. 2019). Similarly, ancestral reconstructions of habitats ignore the potential effect of habitat on diversification. This has been shown to bias ancestral reconstruction analyses (e.g. Maddison Evolution 2006), and models have been developed to account for this, cf all the models from the SSE family, Fitzjohn MEE 2012). I am not suggesting that the authors redo all their analyses (although for some of them it could be quite straightforward), but there

should be at least much more discussion on the limits of the approaches they used / potential consequences for the results / reference to approaches that could be used in the future to deal with these limitations.

(5) The geographical radiation hypothesis presented here is mainly based on the high probability of founder events. Since the model selection when including jump dispersal is highly biased in favor of models incorporating founder events (Conceptual and statistical problems with the DEC+ J model of founder-event speciation and its comparison with DEC via model selection. *Journal of Biogeography*, 45(4), 741-749.), this issue needs to be acknowledged in the discussion as well as the implication for this study.

(6) Other hypothesis of radiation (e.g. adaptive radiation) need to be acknowledged in the discussion as well as the reason why the authors do not favor this hypothesis (despite the radiating group colonizing a new niche – terrestrial environment (showed by the Extended Data Figure 8) – and this radiation been potentially accompanied by adaptation to desiccation and extreme freezing as said in the main text) or both.

Minor comments :

(7) L21. check that “have received less study” is proper English. Maybe “have received less interest”?

(8) L36. Did the authors mean “limited global species richness and wide geographic distributions”? Article 4 claims that microorganisms are ubiquitous.

(9) L61-63. “Microscopy revealed the presence of *P. borealis* in 29% of the samples in the form of dead (13%) or live (16%) cells, confirming its patchy but widespread geographic distribution.” Could it be present but not seen?

(10) L77. “Using rarefaction analyses, we estimated the expected diversity of *P. borealis* for the studied regions to be 415 species on average (Fig. 2, Extended Data Fig. 4).” Extended data 4 (individual-based rarefaction) actually shows a much lower estimation of diversity. Can the authors please correct in the text and explain/interpret the difference between the two estimation results.

(11) L97 and throughout, when reporting diversification rates, please use events per lineage per million year

(12) L133-134 “Evolutionary radiations of relatively young and hyper-diverse clades have long fascinated biologists (25), yet have hardly been documented for micro-organisms (27,28).” See also Martin et al. (*Evolution* 2004) for one of the first studies looking at lineage accumulation curves in microbes, and Morlon et al. (*Evolution* 2012), Louca et al. (*Nature Ecol Evol* 2018).

(13) Fig 2d: Can the authors use “number of reconstructed lineages” instead of “number of species” or something along these lines (the LTT plot represents, at one given time, the number of lineages that left descendants in the present, not the actual number of lineages/species – all the lineages that did not leave extant descendants are missing). Same remark in the caption, “the exponential accumulation of species” is misleading. The LTT plot can give the impression that there is an exponential accumulation of species, but the true number of species can be very different from the LTT plot (species diversity could follow any type of variation through time, including a curve with increasing diversity followed by decreasing diversity).

Martin, A. P., E. K. Costello, A. F. Meyer, D. R. Nemergut, and S. K. Schmidt. 2004. The rate and pattern of cladogenesis in microbes. *Evo-lution* 58:946–955.

Morlon, H., Kempin, B. D., Plotkin, J. B., & Brisson, D. (2012). Explosive radiation of a bacterial species group. *Evolution*, 66(8), 2577-2586.

Moore, B. R., Höhna, S., May, M. R., Rannala, B., & Huelsenbeck, J. P. (2016). Critically evaluating the theory and performance of Bayesian analysis of macroevolutionary mixtures. *Proceedings of the National Academy of Sciences*, 113(34), 9569-9574.

Lewitus, E., Bittner, L., Malviya, S., Bowler, C., & Morlon, H. (2018). Clade-specific diversification dynamics of marine diatoms since the Jurassic. *Nature ecology & evolution*, 2(11), 1715.

Louca, S., Shih, P. M., Pennell, M. W., Fischer, W. W., Parfrey, L. W., & Doebeli, M. (2018). Bacterial diversification through geological time. *Nature ecology & evolution*, 2(9), 1458.

Barido-Sottani, J., Vaughan, T. G., & Stadler, T. (2018). A Multi-State Birth-Death model for Bayesian inference of lineage-specific birth and death rates. *bioRxiv*, 440982.

Maliot, O., Hartig, F., & Morlon, H. (2019). A model with many small shifts for estimating species-specific diversification rates. *Nature ecology & evolution* 3: 1086-1092.

Hoehna, S., Freyman, W. A., Nolen, Z., Huelsenbeck, J., May, M. R., & Moore, B. R. (2019). A Bayesian Approach for Estimating Branch-Specific Speciation and Extinction Rates. *bioRxiv*, 555805.

FitzJohn, R. G. (2012). Diversitree: comparative phylogenetic analyses of diversification in R. *Methods in Ecology and Evolution*, 3(6), 1084-1092.

Maddison, Wayne P. "Confounding asymmetries in evolutionary diversification and character change." *Evolution* 60.8 (2006): 1743-1746.

Editorial Note: This reviewer was invited to provide a brief assessment of a specific technical point.

Reviewer #3:

Remarks to the Author:

This is a fascinating study comprehensively improving our knowledge of the phylogeny and biogeography of a relatively poorly-known group.

I was asked primarily to comment on the BioGeoBEARS biogeography analysis. In general I find it to be of very high quality, and they authors have clearly read the papers on which the models they use are based. In particular, it is great to see the distance-dependent-dispersal model variant ("+x" models) used, as well as Biogeographical Stochastic Mapping to count event types.*

I also reviewed the phylogenetics and diversification sections, methods which I am familiar with. Overall it looks to be very professionally done (without having gone through it in extreme detail). There are various detailed controversies about selection of tree models in phylogenetics, and about diversification inferences from trees of all-living species (as is typical for molecular phylogenies). These issues are well-known, however, and the amount we can do about it is pretty limited, due to computational or theoretical limitations. Of all of the issues involved in applying our imperfect models to a very complex reality, probably the most significant is the fact that extinction is often inferred or assumed to be too low, or zero; this issue is best described by Charles Marshall (2017), *Nat. Eco. Evol.* For the purposes of this paper, I would just make sure there are some caveats described somewhere, and that inferences are thus conditional on the models available.

Minor notes on the BioGeoBEARS analysis:

1.

342 the range evolution model, and the
343 Bayesian Binary Model (BBM) (BAYAREALIKE)69,70.

 technically, the BBM model and the BayArea model are different

* the BBM model just models each area as an independent binary presence-absence character. This was a peculiar invention of the RASP program, where they just threw range data into MrBayes as a

series of independent binary characters. One of several weird features of BBM is that you could have an ancestor of "all zeros" (living nowhere).

* the BayArea model models range as a series of presences/absences, but uses range-expansion / range-contraction events (just like DEC and DIVA) -- so any new area has been colonized from a different, previously occupied area. This allows e.g. distance to be taken into account in dispersal probability, and disallows an "all zeros" ancestor

What BBM and BayArea have in common is that they have a very simple cladogenesis model: the entire range is copied from an ancestor to both descendants. This allows e.g. ancestor range ABCDE to produce 2 daughter species with ranges ABCDE. This is biologically implausible in most cases, but it is computationally fast as it avoids the need for any special cladogenesis calculations.

(BBM and BayArea were both Bayesian in their original implementation)

The BioGeoBEARS "BAYAREALIKE" is an ML implementation of the basic assumptions of BayArea.

Short version, this would make more sense:

and the BayArea model (BAYAREALIKE)

2. Like the diversification & phylogenetics models, lineage extinction is the biggest weakness of the biogeography models. DEC etc. effectively assume a Yule process (pure birth, zero extinction) produced the tree. Matzke (2014) simulations suggest this is not a major issue if extinction is random and speciation > extinction. But, in other situations, it might be.

Note also that incomplete sampling is similar to extinction in effect. And, if there are geographic biases in sampling, this could effect inference.

The most obvious caveat I can see with the sampling here is relative lack of sampling in terrestrial tropics. I know that we cannot just stop science until sampling is perfect, and that permitting, funds, etc. for sampling globally is highly nontrivial. So I would just ask that the authors have a prominent discussion of the possible caveats due to this sampling issue (mostly, you will usually not infer tropical ancestry if tropical tips are not sampled).

Overall I think this paper is very capable, and should be published with minor corrections as above. I do not need to review it again if the above issues are addressed.

Signed,
Nick Matzke

(* Also, I was glad to see no mention of the Ree/Sanmartin critique (2018, J. Biogeography) of the DEC/DEC+J comparison; this critique is fatally flawed in numerous ways, including getting the basic likelihood calculations wrong, as anyone can see for themselves if they run the DEC model on the Ree/Sanmartin example trees; both Ree's Lagrange-DEC and Matzke's BioGeoBEARS-DEC produce the

same likelihood, but they disagree with the numbers reported in the Ree/Sanmartin paper! For more important (!) problems, like the fact that Ree & Sanmartin totally ignored Matzke (2014)'s published simulation tests of the inference, see Klaus & Matzke, 2019, SysBio for a brief overview.)

Below, a point-by-point response to all reviewer suggestions can be found.

Reviewer #1 (Remarks to the Author):

This excellent manuscript represents one of the few phylogeographic studies on soil protists at a global scale. It relies on a broad sampling of a complex of cryptic species of soil diatoms, the *Pinnularia borealis* species complex, and analyses diversification patterns through geological times and across continents. It shows not only that protists have limited geographic distribution areas, but also suggests possible processes involved in speciation.

The manuscript is well-written, and the methods employed are sound and appropriate; this paper will be very influential in the field of protist biogeography and microbial biogeography in general. I would therefore recommend warmly its acceptance. I have only a couple of remarks:

Reply. We would like to thank the reviewer for these interesting comments. We have now revised the manuscript accordingly. More specifically:

- 1) we provided extra information on our environmental metabarcoding approach throughout the manuscript;
- 2) we now used the major biogeographical realms defined by Holt et al. 2013 (*Science*), as a baseline for the geographic regions instead of using latitude to separate regions. The BioGeoBEARS analysis has been rerun to accommodate for these changes;
- 3) we followed the smaller suggestions provided by the reviewer as they added to the quality of the general manuscript.

A more detailed reply for each individual suggestion can be found below.

Line 36 (and also Abstract): In my opinion, the question whether microbes have biogeographies or not is outdated; of course they do! Nowadays, the main question would be what the patterns are and how they are generated. I would remove the part on cosmopolitanism or, at least, give it less importance.

Reply. We agree with the reviewer. For this reason, we omitted reference to microbial biogeography theory from the abstract. We did not remove it from the introduction itself, but added some extra information to give this idea less importance. Specifically:

- Line 36: However, an increasing body of work indicates that micro-organisms instead display distinct biogeographies². Although cosmopolitan species exist^{3,4}, geographically restricted species have been found in marine, freshwater, and terrestrial environments⁵⁻⁸.
- Line 43: Nevertheless, despite recent advances in our understanding of microbial ecology, the nature and drivers of microbial biogeographic patterns and diversification, and how they differ between different groups of micro-organisms, remain poorly resolved, especially for members of the rare biosphere.

Line 46: a detail: It's the eukaryotic tree, the tree of life comprises all three domains of life...

Reply. Agreed. We adjusted this in the text.

Line 50: I would also cite a study concerning soil protists; there are not many, but here is one: Singer, D., Mitchell, E.A.D., Payne, R.J., Blandenier, Q., Duckert, C., Fernandez, L.D., Fournier, B., Hernandez, C.E., Granath, G., Rydin, H., Bragazza, L., Koronatova, N.G., Goia, I., Harris, L.I., Kajukalo, K., Kosakyan, A., Lamentowicz, M., Kosykh, N.P., Vellak, K. & Lara, E. (2019) Dispersal limitations and historical factors determine the biogeography of specialized terrestrial protists. *Mol Ecol*, 28, 3089-3100.

Reply: We agree with the reviewer. The suggested reference has been added to the text (first paragraph).

Line 53: a typo: and, and

Reply. This has been adjusted in the text.

Line 63: please precise that the metabarcoding experiment was based on v4 SSU rRNA. Is this region variable enough to distinguish *P. borealis* spp. from other close related species?

Reply: We thank the reviewer for pointing this out. We now added more detail about the targeted amplicon in the text. The question whether the amplicon is variable enough to distinguish between closely related *P. borealis* species is relevant. In our dataset, the amplicon is variable enough to distinguish between the majority of the species, but not all. To visualize this, we added an extra figure to the supplementary section (Supplementary Fig. 10). In total, 12 species pairs do not differ in their amplicon sequences. Another 51 species pairs differ by only 1-2 bp. For traditional OTU-calling at 97% sequence similarity, this renders the 18S-amplicon not suitable for distinguishing between closely related *P. borealis* species. However, when using an ASV (Amplicon Sequence Variants) pipeline such as DADA2, this problem is partially mediated as ASVs are, in theory, able to distinguish between sequences that differ by 1 bp. Reviewer 2 suggested to include the 18S-metabarcoding dataset to our assessment of the geographic distributions of different *P. borealis* species. Therefore, a more detailed overview of our metabarcoding results can be found in our reply to this remark or reviewer 2 (1-2 C).

Lines 81-82: Are there any clues for claiming that tropical regions would bring a lot of diversity? Do we expect latitudinal gradient patterns of diversity?

Reply. In the original manuscript, we used the tropics as an example of a region that is significantly underrepresented in our study. We did not want to make the claim however that high *P. borealis* diversity is to be expected in the tropics as we currently have no evidence to support this. There are multiple studies that have reported the presence of *P. borealis* in tropical regions in both extant and fossils samples, and our limited sampling on the Mascarene Islands suggests that diversity-levels might be high in some (sub)tropical regions at least. Nevertheless, with this particular statement in our manuscript, we merely wanted to point out that increasing sampling in undersampled regions will most likely increase diversity, and we listed the tropics as one such regions. In order to avoid confusion we removed 'such as in the tropics', so that our statement is more general.

Line 93: As this period is more or less associated with the developing of Poaceae-dominated grasslands, I am wondering if there could be a relationship between Poaceae silica phytoliths and terrestrial diatom frustules... This is just an idea...

Reply. While this is an interesting idea, we found it too speculative to include it in the present manuscript. However, in the revised manuscript we elaborated on the potential link between past environmental change and diversification in the *P. borealis* complex. We included the expansion of grasslands in this paragraph, mostly to highlight the potential increase of available habitat, as *P. borealis* is today often found in open environments. We however did not include the idea on silica phytoliths as we have currently no evidence for such a link. Specifically, we added:

- ➔ Line 311: To date, we can only speculate on whether and how past environmental change might have played a role in the diversification history of *P. borealis* which started around the EOT climate transition. The EOT was characterized by a drop in global temperatures⁵⁰, increased aridification⁵¹, and an increase in seasonality of temperatures and/or precipitation^{52,53}. It is not unlikely that these factors contributed to the dispersal and subsequent diversification of *P. borealis*. Increased aridification is expected to increase the transport of soil particles over vast distances, possibly contributing to long-distance dispersal of soil-dwelling protists. In parallel with the decrease of forest ecosystems throughout the Miocene, grasslands, deserts and tundra vegetation expanded globally, promoted the evolution of large grazing mammals, and likely increased the availability of suitable habitats for *P. borealis*^{54,55}. Today *P. borealis* is commonly found in tundra and desert ecosystems, as well as exposed environments, such as soils and moss vegetation and anthropogenically- and grazer-disturbed sites. Although a direct link between past environmental change and diversification in *P. borealis* cannot be conclusively established, it is conceivable that it has contributed to the accumulation of diversity in this species complex by increasing the availability of spatial opportunity for speciation.

Line 111: Sympatric speciation has also found to be most relevant in another soil protist species complex, the testate amoeba *Hyalosphenia papilio* (Singer et al., 2019). Perhaps authors could discuss this?

Reply. We thank the reviewer for pointing this out. We have now added a reference to the paper by Singer et al. in the manuscript, and more thoroughly discuss the potential role of sympatric speciation in the *P. borealis* complex in the manuscript. More specifically:

- ➔ Line 221. Within-region speciation was at least as important as founder events, as was also observed in soil amoebae⁶.
- ➔ Line 283: Although allopatric species can be ecologically differentiated, in an allopatric setting ecological speciation resulting from local adaptation usually does not play a (key) role in the speciation process itself. Evidence for this stems from the observation that closely-related allopatric species generally occupy highly similar niches, whereas sympatric species tend to show more niche-divergence^{47,48}. Nevertheless, our results do not exclude the possibility of ecological and/or sympatric speciation as within-region speciation also appears to be important in the *P. borealis* complex. Although it is impossible to determine the relative contributions of sympatric, parapatric and allopatric speciation within the vast continental-scale regions including distant islands defined in our analyses, it at least hints towards a potential opportunity for sympatric speciation to occur. To properly assess the role of sympatric and/or ecological speciation in the *P. borealis* complex, detailed ecological trait information should be obtained for all species, and, if possible, complemented by comparative/population genomic analyses. Nonetheless, it is highly unlikely that sympatric speciation on its own has generated the high levels of diversity of *P. borealis* observed in this study, nor that divergent selection was solely responsible for speciation after the establishment of physical barriers to gene flow. Furthermore, as noted above, it is not unlikely that different drivers are acting together in shaping the extreme species-level diversity observed in *P. borealis*.

Figure 1: I suggest presenting a map with the classical biogeographic realms (Neotropical, Palaeotropical, etc...). Or at least a map with realistic climatic zones that stick better to reality would make sense here. Figure 2: Same here, I would divide into biogeographic realms.

Reply. Following this suggesting, we redefined the geographic zones on the map. Instead of using latitude as a defining character, we now used the major biogeographical realms as defined by Holt et al. 2013 (*Science*), including two extra regions: Antarctica and Sub-Antarctica. This implied that two samples in northern North America had to be changed from geographic region. Apart from this, all samples remained in their originally defined areas. To accommodate this in the historical biogeography analysis, the BioGeoBEARS analysis was rerun, using the adjusted geographic distributions, and, where necessary, adjusted geographic distances between regions. The geographic distributions have also been updated in every figure in which these data were incorporated. As in our previous analysis, we kept the samples from Amsterdam Island and Saint-Paul island in one geographic region with Tasmania. These islands were not attributed to any biogeographical zone by Holt et al. 2013. Given the fact that we could not introduce them as a separate region as our dataset only allowed a maximum of eight geographic regions due to computation limitations, we merged these samples with an already existing region. The fact that Tasmania has the same climatological conditions as Amsterdam and Saint-Paul island (Köppen-Geiger climate classification: class Cfb, a mild oceanic climate), and given that they are located on similar latitudes, justifies our choice to merge these samples within a single region. We would also like to point out that at least one *P. borealis* species occurs in both Amsterdam/Saint-Paul island and Tasmania, further strengthening evidence for affinity between these regions for the *P. borealis* complex.

In addition to changes in the biogeographic zones, the new BioGeoBEARS analysis also incorporates knowledge obtained from the 18S-metabarcoding analysis. As outlined above, this implied that one *P. borealis* lineage (code (Tor12)d) has been given a wider geographic distribution occurring in South America and Continental/Maritime Antarctica. This was done because a 18S-ASV that is identical to the 18S-haplotype characterizing *P. borealis* lineage (Tor12)d, was found by metabarcoding in a sample of Deception Island (South Shetland Islands, Maritime Antarctica). We refer to our reply (1-2 C) to one of the remarks of reviewer 2 for additional information on this matter.

Reviewer #2 (Remarks to the Author):

This paper aims at characterizing the diversification dynamic of a soil diatoms species complex and understand the process generating this diversification dynamic. To do so, the authors (1) cultivated diatom strains from 207 samples collected worldwide (1) delimited species thanks to 2 fast evolving loci (2) estimated the expected global diversity of the group using a rarefaction analyses (3) reconstructed a time-calibrated phylogenetic tree using 6 markers and fossils (4) analyzed the diatom species complex diversification dynamics with LTT plots and phylogenetic diversification models and (5) analyzed the ancestral biogeography of the group. The authors conclude (1) that the studied diatom group is exceptionally diverse, with high diversification rates compared to other diatom groups that would be linked to a radiation of the group after the Oligocene-Eocene cooling event and (2) that the group has a patchy distribution and is characterized by frequent colonization of novel geographical areas, emphasizing the role of allopatric speciation and putting forward the geographical radiation hypothesis.

The originality of the study lies in the analysis of diversification dynamics of terrestrial protists often omitted of microorganisms surveys. The overall message of the paper - showing how rare species complex with high species diversity can undergo geographical radiation and thus speciate in allopatry mainly, is original regarding the current literature. The paper represents a significant amount of work, is clearly written, and the results are interesting. Our main concern is that the sampling technique (cultivation) and effort (207 samples worldwide) might be responsible for the conclusions drawn by the authors, and that this possibility is not sufficiently addressed. There are other major limitations of the approaches used (detailed below) that are not even mentioned by the authors. In conclusion, we think that the paper can be a nice addition to the field if the authors can be more convincing that their results are clearly supported. Some of the conclusions will probably need to be toned-down to avoid over-statement.

Reply: We would like to thank the reviewer for these valuable comments. We have now revised the manuscript accordingly, and believe this has greatly improved the manuscript. More specifically:

- 1) we provided additional information on sampling strategy and effort to the manuscript, as well as to the Reporting Summary that is associated with the manuscript;
- 2) we added a more detailed analysis of the environmental metabarcoding dataset to the manuscript, and incorporated the metabarcoding results in the assessment of the geographic distributions of individual species used as input for the BioGeoBEARS analysis;
- 3) we performed three additional analyses on diversification rate reconstruction, using the BD-model in the R-package Geiger, the Missing State Speciation and Extinction model (MiSSE), and the Cladogenetic Diversification rate Shift model (ClADS). The results of all three models were added to the manuscript;
- 4) we expanded our discussion on evolutionary radiations, and diversification in the *P. borealis* species complex;
- 5) we discussed potential shortcomings of some of the methods used in relation to the results of our analyses;
- 6) we added an extra analysis on population differentiation and phylogeography of the most widely distributed and common *P. borealis* species recovered in our study, providing additional evidence for molecular differentiation in geographic isolation.

A more detailed reply for each individual suggestion can be found below. In points (1) and (2), the reviewer asked multiple questions that are related to each other. Below, we will answer these questions/remarks in several parts (A – B – C):

(1-2) A) It could be that the patchiness of *P. borealis* geographic distribution comes from the sampling technique and effort. Can the authors provide justification of if/why the cultivates from the samples provide a good estimate of diatom diversity in each sample? If a strain is not cultivated in a sample, how confident can we be that it is not present? If a strain is not observed in one of the few samples representing a large biogeographic region, how confident can we be that it is not present in that geographic region? The authors could for example use sub-sampling to assess the robustness of their ancestral biogeographic reconstructions to sampling effort.

Reply. Although we provided the most extensive sampling of a diatom species complex to date, and to the best of our knowledge, also the most extensive sampling of any terrestrial protist, the reviewer is correct in pointing out that our sampling effort was not complete (as also shown by our species accumulation curves). Nevertheless, we believe our sampling was sufficiently complete to allow for the analyses in our study as supported by several observations.

First, we would like to refer to the individual-based rarefaction and extrapolation curve shown in the supplementary section of the manuscript (Supplementary Fig. 4). This curve displays the sampled and expected *P. borealis* diversity within the investigated samples. The sampled diversity comes down to 126 species, whereas the expected diversity equals 143 species (126.9 – 160: 95% confidence interval). Hence, the difference between the sampled and expected diversity is thus relatively small, indicating that in the entire dataset, we found the majority of the *P. borealis* species that are present in a sample. This information has now also been added to the manuscript. Specifically:

- Line 111: These additional species are predominantly to be expected when additional samples are investigated, as individual-based rarefaction analyses indicated that within the set of investigated samples, the overall majority of *P. borealis* species present has been found (Supplementary Fig. 4).

Second, the generally high number of samples per geographic region, combined with the broad definition of geographic areas in our study, ensured that we obtained a fairly good coverage of the diversity in these regions. However, we admit that it can never be excluded that some species have wider geographic distributions than here indicated, and additional sampling might reveal that some species are present in areas in which they were not observed in our dataset. This issue cannot be simply tackled by ‘gathering more samples’. In fact, the sampling effort needed to fully sample the *P. borealis* complex might be impractically high, as extrapolation of our species accumulation curves suggested that *P. borealis* strains would need to be cultured from over 1,500 environmental samples before the species accumulation curve starts levelling off. If our success rate of finding *P. borealis* is extrapolated, it becomes clear that ca. 9,500 environmental samples would need to be gathered to find the majority of *P. borealis* lineages. This in itself, is not feasible. This has now also been made more clear in the manuscript. Specifically:

- Line 114: Although our survey offers, to the best of our knowledge, the most comprehensive sampling of any diatom complex to date, and of protists in general, it is clear that extending sampling into additional regions would substantially increase the global number of known *P. borealis* species. Indeed, extrapolation of our species accumulation curves suggests that *P. borealis* strains would need to be obtained from over 1,500 environmental samples before the species accumulation curve starts levelling off. Given our success rate of finding *P. borealis* in

a given sample, this would imply that ca. 9,500 environmental samples would need to be gathered to find the majority of *P. borealis* lineages.

Third, an independent, albeit more limited, assessment of *P. borealis* biogeography was provided by our environmental metabarcoding analysis of the 18S gene and gave almost identical results as our culture-based assessment. In the original manuscript we did not elaborate on the metabarcoding analysis, but in a response to another suggestion of reviewer 2, we added a more comprehensive analysis of these metabarcoding data to the paper. An overview of these results can be found in response to question (1-2 C) of reviewer 2.

To accommodate the impractically high sampling effort needed to fully sample our protist species complex under study, the reviewer suggested that sub-sampling could be used to assess the potential impact of false absences in our dataset. We find this an interesting idea, but would like to point out that sub-sampling would require repeated analysis of historical biogeography in BioGeoBEARS. Unfortunately, these analyses are computationally highly intensive and time-consuming, and for this reason not feasible to repeat x-number of times to assess the robustness of our results in a statistically sound way. Probably for this same reason, we could not find any precedent in the literature of a study that has done exactly this. Nevertheless, we agree with the reviewer that it is necessary to give more attention to these matters in our manuscript, and we have now done so. Specifically:

- Line 206: Using the culture and metabarcoding data to assess species distributions, we found the majority of examined species to be relatively restricted in their geographic distributions (Fig. 2). However, we also recovered two species that occur in both hemispheres (Fig. 2). This is in concordance with previous results on freshwater protists that showed that closely-related species can have geographically restricted as well as cosmopolitan/bipolar distributions^{5,16}. Nevertheless, despite our unprecedented extensive sampling effort, in which more than 1,500 environmental samples were investigated, large portions of the planet remain unsampled, particularly (sub-)tropical and temperate regions. The species distributions uncovered in our study thus have to be seen as approximations, and it is not unlikely that some of the species that now have seemingly restricted distributions might be geographically more widespread.
- Line 240: In light of these results, it has to be noted that our extensive but still incomplete geographic sampling might have affected our results on the historical biogeographical processes shaping the evolutionary history of the *P. borealis* complex, as anagenetic and sympatric processes are likely to be of higher importance in a scenario with wider species distributions. Such wider species distributions may be expected to be uncovered when additional samples/regions would be investigated. However, additional sampling is also likely to uncover additional (rare) species-level diversity with restricted distributions. Furthermore, absence of unsampled species-level diversity in the *P. borealis* phylogeny may have influenced the inference of ancestral ranges, which could in turn have impacted our estimates on historical biogeographical processes. Nevertheless, evidence for a geographic factor in the diversification of *P. borealis* was also uncovered when investigating intraspecific diversity in one of the *P. borealis* species.

(1-2) B) Are diatoms in really low abundances easily cultivated?

Reply. This is an interesting question. We will answer this question for *P. borealis* specifically, not for diatoms in general. *Pinnularia borealis* was carefully chosen for this study for multiple reasons. It has a highly characteristic morphology, and for this reason, is easy recognizable at low magnifications in a light microscope, and as a consequence this makes cell isolation substantially easier. Although they are slow growers, *P. borealis* species survive well in culture. In fact, they can survive up to one year in culture without medium refreshment (Pinseel, own observations), and can be successfully cryopreserved without cryoprotectants (see also Stock et al. 2018 and Hejduková et al. 2019). Furthermore, *P. borealis* survives prolonged periods in sampling recipients, such as sampling bags and falcon tubes, and for this reason is easily transported from the field to the lab (Pinseel, own observations). Altogether, these

characteristics ensure that *P. borealis* is an ideal choice to study rare biosphere diatom species in culture. This has now been made more clear in the text. Specifically:

- Line 66: Although seldom locally abundant, members of the complex are easily recognizable at low magnifications in light microscopy, can survive prolonged periods in suboptimal conditions including sampling recipients, and although they are generally slow growers, they are easy to maintain in culture²².

Although *P. borealis* cells have low abundances in the overall majority of the environmental samples in which it has been found, this was accommodated by careful sample treatment by the first author. All environmental samples were subsampled in multiple wells of 12-well plates. In doing so, care was taken to take material from different parts of the sample, and if the sample was heterogeneous (for example, a mix of soil and moss), multiple subsamples from these different parts were taken. These samples were subsequently screened in a light microscope over a course of multiple weeks to months. In case only dead valves of *P. borealis* were observed, samples were screened over longer time periods (multiple months). Although time-consuming, this approach ensured that the chances of observing living *P. borealis* cells were maximized. In conclusion, when taking sufficient time, and using appropriate sampling protocols, *P. borealis* is easily cultivated. The following information has now also been added to the Supplementary Methods:

- Upon arrival in the laboratory, small quantities of the natural material (subsamples) were incubated for several weeks to months in WC medium¹, without pH adjustment or vitamin addition, at 4 °C (for polar and temperate regions) or 18 °C (for subtropical regions), 5 – 10 μmol photons m⁻² s⁻¹ and a 12:12h (light:dark) cycle. Although the abundances of *P. borealis* cells were low in the overall majority of the samples, this was accommodated by careful sample treatment. All environmental samples were subsampled in multiple wells of 12-well plates. In doing so, care was taken to take material from different parts of the sample, and if the sample was heterogeneous (for example, a mix of soil and moss), multiple subsamples from these different parts were taken. These samples were subsequently screened repeatedly in a light microscope over a course of several weeks. In case only dead valves of *P. borealis* were observed, samples were screened over longer time periods (up to four to six months). Although time-consuming, this approach ensured that the chances of observing living *P. borealis* cells were maximized.

(1-2) C) Maybe the metabarcoding dataset can be used to test/validate the completeness of the diversity recovered in each sample.

Reply. Following this very relevant suggestion, we added a more detailed analysis of the 18S-ASVs (Amplicon Sequence Variants) belonging to the *P. borealis* species complex that were recovered in our metabarcoding analysis (see Supplementary Fig. 10 for a detailed overview). We first would like to point out that our 18S-amplicon has near species-level resolution: the majority of the *P. borealis* species are clearly differentiated for the 18S-amplicon, but there are twelve species pairs that do not differ, or only differ by 1-2 base pairs (see also Supplementary Figs 10d – 10e). This renders the 18S-amplicon insufficient to distinguish between closely related *P. borealis* species when traditional OTU-calling at 97% sequence similarity is applied. However, when using an ASV pipeline such as DADA2, this problem is partially mediated as ASVs are able to distinguish between sequences that differ by 1 bp. This means that the majority of the *P. borealis* species in our dataset can be identified using the 18S-amplicon. Therefore, we assigned identifications to each *P. borealis* ASV using Mothur. All ASVs were subsequently aligned to their most closely related reference sequence to check sequence similarity. In many cases (see also Supplementary Fig. 10c), these ASVs were found to be identical to reference sequences of *P. borealis* obtained via Sanger sequencing. We then used the Mothur-IDs to check for the occurrences of different species in different samples (see next

paragraph). Whenever ASVs belonged to species pairs that could not be distinguished by means of the 18S-amplicon, the combined geographic distribution of both species as assessed by the culture data was taken into account to assess congruence between the culture and metabarcoding datasets, ensuring a conservative approach.

In total, we analyzed 132 environmental samples via environmental metabarcoding. In all these samples, the presence of *P. borealis* was visually confirmed. From 49 samples we could retrieve *P. borealis* 18S-ASVs using metabarcoding. For 32 of these, culture data were available, allowing a direct comparison between the culture data and the metabarcoding data (see Supplementary Fig. 10a). In this figure, the number of species per sample (culture vs. metabarcoding data) is indicated. For the metabarcoding dataset, the different colors indicate whether the species retrieved by metabarcoding were (i) also found in the culture dataset of the same sample (blue), (ii) not found in the culture dataset of the same sample, BUT retrieved in cultures of other samples from the same geographic region (yellow), or (iii) not retrieved in cultures of other samples from the same geographic region (red). It becomes immediately clear from Supplementary Fig. 10a that all but one sample contained *P. borealis* lineages that were also detected by culture data in the same sample, or that were detected in other samples from the same geographic region, and that in the majority of the samples, metabarcoding finds the same species as in the culture dataset. In many cases the retrieved ASVs were identical to the reference sequences of the species in question. This indicates that our culture effort of the individual samples was relatively complete. This also confirms the results of the individual-based rarefaction curves.

Upon investigating the 18S-ASVs more closely, two environmental samples contained ASVs that were not found in the same geographic region using culturing (indicated in red in Supplementary Figs 10a – 10c). One of these ASVs (ASV_4606) is identical to a 18S reference sequence of a *P. borealis* lineages that was until now only recovered from Patagonia (*P. borealis* lineage (Tor12)d). ASV_4606 was found in a sample from Antarctica. The other ASV (ASV_5117) differs 2 bp with two *P. borealis* lineages from the (Sub-)Antarctic region that have identical 18S-sequences for the amplicon (lineages KGI16_03_16 and MI16_20_19). ASV_5117 was found in Greenland. Given that a substantial number of species differs by 2 bp or less, it is unlikely that ASV_5117 is conspecific with KGI16_03_16 or MI16_20_19. It could, for example, represent an as yet unknown *P. borealis* species that is closely related to the latter strains. Indeed, several other ASVs differ with 1 to 6 bp with their most closely related reference sequences (Supplementary Fig. 10c). These ASVs can either represent intraspecific variation within known *P. borealis* species, or they represent yet unknown species-level diversity. These findings support our claim that with additional sampling, additional diversity is to be found. However, given the relatively low resolution of the 18S-amplicon, it is difficult to make strong conclusions based on these data.

Finally, the finding of *P. borealis* lineage (Tor12)d in Antarctica, motivated us to rerun the BioGeoBEARS analysis, now including Antarctica for the distribution of (Tor12)d. All figures of the BioGeoBEARS analysis have been updated accordingly. For all other *P. borealis* species, inclusion of the metabarcoding data did not change the geographic distribution of individual species. However, to avoid potential false identifications, we only took into account ASVs that were identical to reference sequences. ASVs that differed with their most closely related reference sequence were not taken into account to determine the distribution of individual *P. borealis* species.

A paragraph that discusses the metabarcoding results has now also been added to the manuscript. Specifically:

- ➔ Line 182: In order to account for incomplete sampling in the culture dataset, we combined the cultures and metabarcoding data (18S) to determine the geographic distributions of individual *P. borealis* species, and to assess congruence between both datasets. The metabarcoding dataset was analyzed using the R-based pipeline DADA2⁴¹, which generates Amplicon Sequence Variants (ASVs). In contrast to Operational Taxonomic Units (OTUs), which cluster

multiple sequences together, ASVs represent unique haplotypes. This makes it possible to distinguish sequences with as little as one base pair difference. The 18S gene has limited species-level resolution in the *P. borealis* complex, as several species pairs differ by only one to three base pairs, and twelve species pairs showed identical 18S-haplotypes (Supplementary Fig. 10). Therefore, only ASVs that were identical to reference 18S-sequences that were obtained by Sanger sequencing (Supplementary Table 2) were taken into account to assess distributions of individual *P. borealis* species. In case ASVs belonged to species pairs that could not be distinguished by means of the 18S-amplicon, the combined geographic distribution of both species as assessed by culture data was taken into account to assess congruence between the culture and metabarcoding dataset. In general, the geographic distributions (following the classification in Fig. 1) of individual species obtained by metabarcoding and culture data were identical, with the exception of one *P. borealis* species that was found to inhabit an additional region in the metabarcoding dataset (Supplementary Fig. 10). Several 18S-haplotypes retrieved by metabarcoding differed by one to six base pairs from their most similar reference sequence (Supplementary Fig. 10). These haplotypes could represent intraspecific or intragenomic variation in 18S (see also ref.⁴²). Alternatively, (part of) these unknown 18S-haplotypes could represent as yet unknown species-level diversity in the *P. borealis* complex, confirming the results of our species-accumulation curves that suggest that additional species-level diversity would be uncovered with additional sampling.

(3) One of the main results put forward by the authors is that *P. borealis* shows extraordinarily high species-level diversity, and elevated diversification rates, compared to other diatom groups. This is based on comparisons of estimates of species richness and diversification rates from Ref 12. However, a recent paper by Lewitus et al. (Nature Ecol Evol 2018) reports diversification rates for diatoms as a whole and various diatom groups that are similar to the rates reported here (see their Fig 1b & 3b). Some clades even have much higher diversification rates.

Reply. We would like to thank the reviewer for bringing the paper by Lewitus et al. to our attention. We were aware of this study, but originally did not refer to it as we believe that their inferred diversification rates are not directly comparable to ours due to several reasons:

(1) The phylogenetic tree of Lewitus et al. is based on a single gene (18S), and the overall majority of the sequences in their phylogeny originate from 18S-OTUs obtained from environmental metabarcoding datasets of the marine realm. This means that the majority of the sequences in the phylogeny are only a few hundred base pairs long. While building a species tree of the entire diatom clade based on such a short DNA fragment of a single gene has provided important insights into the effect of past environmental changes on the evolution of diatoms, it is less suitable to provide a backbone phylogeny that is sufficiently robust to serve as input to estimate absolute diversification rates for diatoms. This is demonstrated in Fig. 1a in Lewitus et al. The topology of the species tree of the diatoms as retrieved by Lewitus et al. does not fit in any previously published work on the diatom phylogeny (see amongst others Theriot et al. 2010 *Plant Ecology & Evolution*, Parks et al. 2018 *Mol. Biol. Evol.*, and Nakov et al. 2018 *New Phytologist*). For example, in the phylogeny of Lewitus et al. the raphid pennate diatoms are shown to be polyphyletic, as they consist of five independent clades, while the raphid pennate diatom clade has long been known to be monophyletic. In addition, the araphid pennate diatoms turn out the most diverse group in the study of Lewitus et al., whereas it is generally known that the raphid pennate diatoms are by far the most diverse diatom clade (see also Nakov et al. *New Phytologist*). Furthermore, the phylogeny by Lewitus et al. shows relatively few deep branches, and a large number of short branches, which is not conform other studies that used multiple genes to reconstruct the diatom phylogeny. Possibly, this could be a byproduct of using short DNA-fragments in FastTree to build a phylogenetic tree. Importantly, if many of the branch lengths in the tree of Lewitus et al. are indeed too short, this could have had resulted in an overestimation of the inferred diversification rates.

(2) The phylogenetic study of Lewitus et al. was based on individual 18S-OTUs obtained by environmental metabarcoding. These OTUs do not necessarily represent individual species as a single OTU might harbor multiple species, and a single species can be scattered over multiple OTUs. Furthermore, intragenomic variation in 18S-haplotype diversity is estimated to be high in diatoms (see for example Alverson et al. 2007 *J. Phycol.* or Godhe et al. 2008 *Appl. Env. Microbiol.*), and might inflate the diversity of 18S-haplotypes in environmental metabarcoding studies. In other words: Lewitus et al. estimated diversification rate as a function of the rate at which new 18S-OTUs evolve, which is unlikely to be the rate at which new species evolve.

Together, points (1), and (2) suggest that it is difficult to directly compare the rates obtained by Lewitus et al. with our multi-gene based study. Instead, the rates obtained by Nakov et al. 2018 *New Phytol.* are probably better suited for such comparison given that these authors used species as individual entities for their analysis, taking unsampled diversity into account, and used a strongly supported multi-gene backbone phylogeny of the diatom clade.

Nevertheless, we do not think that our manuscript is the correct place to discuss the conclusions of Lewitus et al. We therefore opt to not explicitly refer to the rates obtained by Lewitus et al., although we added some extra sentences to the paragraph dealing with diversification rates and citing Lewitus et al. so that future readers of our manuscript know we have taken this study into account. In addition, we added some extra information on the results obtained by Nakov et al., since in the original manuscript we only referred to Nakov's rates of the diatom clade as a whole. In the supplementary material, Nakov et al. gives an overview of clade specific diversification rates, showing that some clades evolve at rates that are similar to those of *P. borealis*. Most interestingly, Nakov's modelled rates of the pinnularoid diatoms, to which *P. borealis* belongs, are well below the estimates for diversification rates in *P. borealis*, suggesting that *P. borealis* is evolving at faster rate than the pinnularoids as a whole. This information has now also been added to the manuscript. Specifically:

- Line 162: In addition, the retrieved rates are in line with those observed in macro-organismal radiations^{35,36}, and well above estimates over the diatom tree as a whole, which maximally equaled $\sim 0.06 \text{ lineage}^{-1} \text{ Myr}^{-1}$ ¹⁸. However, several specific diatom clades have been found to evolve at rates that are similar to, or even exceed those of *P. borealis*^{18,37}. For example, net diversification rates between ~ 0.075 and $\sim 0.20 \text{ lineage}^{-1} \text{ Myr}^{-1}$ were reported for the eucoconeid, eunotoid and cymbelloid diatoms¹⁸. Previous estimates for the pinnularoid diatoms as a whole, to which *P. borealis* belongs, maximally equaled $\sim 0.075 \text{ lineage}^{-1}$ ¹⁸, suggesting that the *P. borealis* complex is evolving at a faster rate than its closest sister lineages. However, more complete taxon sampling of the entire *Pinnularia* tree as well as of other major diatom lineages is needed to gain more robust estimates of their diversification rates in order to improve the comparison with *P. borealis*.

(4) Some of the diversification results are based on BAMM analyses. BAMM has been heavily criticized (e.g. Moore et al. PNAS 2016) and alternatives have been developed (Barido-Sottani et al. BioRxiv 2018, Maliet et al. Nature Ecol Evol 2019, Hohna et al. 2019). Similarly, ancestral reconstructions of habitats ignore the potential effect of habitat on diversification. This has been shown to bias ancestral reconstruction analyses (e.g. Maddison Evolution 2006), and models have been developed to account for this, cf all the models from the SSE family, Fitzjohn MEE 2012). I am not suggesting that the authors redo all their analyses (although for some of them it could be quite straightforward), but there should be at least much more discussion on the limits of the approaches they used / potential consequences for the results / reference to approaches that could be used in the future to deal with these limitations.

Reply. We thank the reviewer for this comment. Indeed, we are aware that BAMM has been criticized over the years. For this reason, our original manuscript also included a CoMET analysis in TESS to have two independent estimates for net diversification rates and diversification rate

shifts. It was our idea that if both analyses gave similar results (which is the case), the inferred rates should be reliable, given the limitations of both approaches. However, this comment of the reviewer prompted us to run additional analyses on diversification rates in *P. borealis* to test the robustness of our original results. More specifically, we chose to run three additional models. First, we ran the BD model using the R-package Geiger. This is a very simple function that allows the calculation of net diversification rates of a clade, given a fixed parameter for relative extinction rates. Since the extinction rates of *P. borealis* are not known *a priori*, we used the diatom-wide extinction rate obtained in a previous study (Nakov et al. 2018 *New Phytologist*). Second, we ran the MiSSE model. MiSSE is similar to HiSSE but does not require trait information. Third, we followed the suggestion of the reviewer, and chose to run ClaDS which is a model that allows to detect small shifts in diversification rates, in contrast with to the large diversification rate shifts which are typically inferred by models such as BAMM. We originally ran ClaDS2, but due to convergence problems, limited the final analysis to ClaDS0 for the 100% sampling fraction. Convergence problems in analyses that include extinction (such as ClaDS2) and/or unsampled diversity, are well-documented in the ClaDS paper. In addition, we contacted the first author of the ClaDS paper to obtain more information on this matter. More specifically, we received the following information: “*However this version of the model (the author is here referring to ClaDS2) is very long to run even for moderate size trees, and in empirical trees epsilon is often estimated to be very close to 0, so that the estimates of ClaDS0 generally are a good approximation of what you would obtain by fitting this model.*”

The diversification rates that were inferred by all three additional models were directly in line with the rates observed by BAMM and TESS, although both MiSSE and ClaDS showed larger variations in inferred diversification rates throughout the phylogeny, with generally decreasing rates through time. They thus do not change the general conclusions of our study. We added the results of the newly run models to the manuscript. Specifically:

- ➔ Line 149: In order to take unsampled diversity into account three sampling fractions were used, assuming complete and two scenarios of incomplete sampling. Using the approach by Magallon & Saunders³¹, we recovered net diversification rates of 0.11 (complete sampling fraction), 0.15 (30% sampling fraction) and 0.18 (10% sampling fraction) events per lineage per million year ($\text{lineage}^{-1} \text{Myr}^{-1}$). These rates were very similar to those recovered by BAMM³² (Supplementary Fig. 7). TESS²⁹ recovered slightly higher net diversification rates ranging between ~0.15 and ~0.30 $\text{lineage}^{-1} \text{Myr}^{-1}$ when taking unsampled species diversity into account (Supplementary Fig. 7). The MiSSE³³ model recovered more variability compared to BAMM, with diversification rates ranging between ~0.089 and ~0.22 $\text{lineage}^{-1} \text{Myr}^{-1}$ for the 30% and 10% sampling fractions (Supplementary Fig. 8). Whereas BAMM and TESS did not detect evidence for significant rate shifts throughout the evolutionary history of *P. borealis*, MiSSE suggested that net diversification rates declined through time, and ClaDS³⁴ retrieved several small shifts in diversification rate throughout the phylogeny, generally resulting in a declining diversification rate through time (Supplementary Fig. 8). Despite these variations between the different models, the modelled net diversification rates were highly similar across models.
- ➔ Line 530: Net diversification rates were calculated using the *bd.ms* function in the R-package Geiger⁸⁸, based on ref.³¹. This function requires an extinction rate as input. Since no information is available on extinction rates in the *P. borealis* complex, we used the diatom-wide relative extinction rate obtained by ref.¹⁸ as input: 0.751 $\text{lineage}^{-1} \text{Myr}^{-1}$. The *bd.ms* function also requires an estimate of the total number of species within the dataset. Based on the extrapolation of the rarefaction curves, the average number of sampled species was estimated on 30.36%. This is likely still an underestimating on a global scale, resulting from the lack of data from large geographic areas. We therefore also ran *bd.ms* with a sampling coverage of 10%. Finally, we considered a complete sampling coverage to provide an absolute baseline for *P. borealis* diversification.
- ➔ Line 559: The MiSSE model was run using the R-package hisse³³. For each sampling fraction, four analyses were run, using one to four hidden states for the turnover fraction. The extinction fraction was not varied. All other settings were kept default. The AIC (Akaike Information Criterion) values of these models differed only with a couple of units. Therefore, the results of the four models were averaged per sampling fraction to obtain a more robust estimation of diversification rates.

→ Line 565: At last, ClaDS is able to detect small rate shifts across a phylogenetic tree³⁴. ClaDS allows to run several models, assuming extinction rates that are negligible (ClaDS0), homogeneous across all lineages (ClaDS1), or varying across lineages, but with a constant turnover (ClaDS2). We initially ran ClaDS2 using the R-package RPANDA⁹¹ for all three sampling fractions for 500,000 generations using three chains (~3 weeks run time per analysis), after which all analyses were still far from converged. These computational limitations and convergence problems when extinction is taken into account and/or missing taxa are present in the tree are well-described in the ClaDS paper³⁴. Therefore, we limited the analysis to a ClaDS0 run assuming a 100% sampling fraction. The results of ClaDS thus have to be seen as an illustration of the prevalence of small rate shifts throughout the history of *P. borealis*, rather than absolute values of *P. borealis* diversification. We ran ClaDS0 initially for 2,000,000 iterations, thinning every 200,000 iterations and using three chains. Subsequently, the gelman statistic was calculated and additional rounds of 2,000,000 iterations were added until the gelman factor was below 1.05. Following visual inspection of the MCMC chains, the first 900 recorded chain states were discarded, after which the posterior maximas of the rates were calculated.

We also agree with the reviewer regarding the comment on the ancestral habitat reconstructions. However, in our study we only inferred diversification rates of the *P. borealis* complex, and did not estimate rates in the *Pinnularia* phylogeny as a whole. We chose to not include the sister groups of *P. borealis* in our diversification rate analyses as we believe that these clades have been insufficiently sampled to allow for detailed analyses of diversification rate shifts within the *Pinnularia* tree. Depending on the source, this genus is estimated to have ca. 700-1500 representatives, while for less than 50 of these species sequences are available on GenBank. For this reason, including a model with trait variation (such as the majority of SSE models) is not feasible in our study setting (note that the majority of the *P. borealis* species in our dataset inhabit the same environment). Instead, we chose an analysis in Mesquite to visualize the shift in habitat type that has been at the root of the *P. borealis* complex.

We agree with the reviewer that our study would benefit from additional discussion on the limits of the approaches that we used in our analyses. For this reason, an extra paragraph has been added to the end of the manuscript. Specifically:

→ Line 141: The fossil record of *P. borealis* is rudimentary, and its uniform morphology²² precludes direct inference of species turnover rates. Therefore, diversification rate analyses using various models were used to assess diversification rates in the complex based on the time-calibrated phylogeny. Estimating speciation and extinction rates from a phylogeny which only contains extant diversity is a difficult task which is sensitive to undersampling²⁹, and is especially prone to underestimating extinction rates³⁰. As a consequence, the inferred diversification rate estimates are subject to the shortcomings of the chosen models. To, at least partially, alleviate this issue, we chose to adopt a suite of diversification rate models, and assessed congruence amongst the results.

(5) The geographical radiation hypothesis presented here is mainly based on the high probability of founder events. Since the model selection when including jump dispersal is highly biased in favor of models incorporating founder events (Conceptual and statistical problems with the DEC+ J model of founder-event speciation and its comparison with DEC via model selection. *Journal of Biogeography*, 45(4), 741-749.), this issue needs to be acknowledged in the discussion as well as the implication for this study.

Reply. We thank the reviewer for this comment. We are aware of the critiques that have been raised concerning the DEC+J model. This is still an ongoing debate, as critiques have also been raised on the paper by Ree & Sanmartin 2018 (*Journal of Biogeography*) – see also the last comment by reviewer 3. To address the remark of the reviewer, we added a sentence on potential flaws of the DEC+J model, referencing both sides of the debate. In addition, we also ran the analyses on Biogeographical Stochastic Mapping on the second and third best models in our study (DIVALIKE+J+x, and BAYAREALIKE+J+x), and added these results to the

supplementary section of the paper. In these models as well, founder events are inferred to be highly important in the *P. borealis* species complex, underscoring our results. Specifically:

- Line 586: It is worth noting that critiques have been raised on the use of the founder-event parameter j in DEC models⁹², although others have argued that the latter study shows several fundamental flaws⁹³.

We would also like to emphasize that the high probability of founder events is not the only evidence we put forward regarding the role of geographic isolation in speciation in the *P. borealis* species complex. Using the +x models in BioGeoBEARS allowed assessing whether a distance effect was present in our dataset, and this was indeed found to be the case. In other words: dispersal is not unlimited in *P. borealis*, and this in itself is direct evidence that conditions for allopatric speciation are likely to occur. In addition, we performed an additional analysis on one of the *P. borealis* species in our dataset (JRI15_10_06). This species is the most abundant and widespread species in our dataset, and was found in both hemispheres. Using Analysis of Molecular Variance (AMOVA), we showed that this species is geographically structured between the Northern and Southern Hemisphere. Molecular time-calibrated phylogenetic analysis of different haplotypes within this species revealed that differentiation between the Northern and Southern Hemisphere most likely occurred during the Pleistocene glacial-interglacial cycles. Such scenario can be most easily explained by increased geographic isolation between distant populations, such as which could occur during glacial cycles. We have now also added these extra analyses, including figures, to the manuscript. Specifically:

- Line 249: Nevertheless, evidence for a geographic factor in the diversification of *P. borealis* was also uncovered when investigating intraspecific diversity in one of the *P. borealis* species. It concerns the most widely distributed *P. borealis* species in this study, referred to by its reference strain JRI15_10_06 (clade 1). It was observed in 30% of the samples with culture material, represented 18.6% of all established cultures, and was also the most common species in the metabarcoding dataset. In addition, this species showed substantial sequence variability in the *cox1*-gene. Analysis of Molecular Variance (AMOVA)⁴⁴ revealed that this molecular variation was geographically structured between the Northern and Southern Hemisphere (Supplementary Fig. 13). Molecular time-calibrated phylogenetic analysis further showed that this *P. borealis* species comprises three distinct clades (one in the Southern Hemisphere, and two in the Northern Hemisphere) that diverged during the Pleistocene, a period characterized by repeated glacial-interglacial cycles⁴⁵ (Fig. 4, Supplementary Fig. 13). Whereas no haplotypes are shared between the Southern and Northern Hemisphere, members of the northern clades occur sympatrically in the same regions (Fig 4., Supplementary Fig. 13). These observations are concordant with geographic isolation during glacial maxima, for example in glacial refugia or nunataks, and range expansion during interglacials (a vicariance scenario), or long-distance dispersal between hemispheres followed by local divergence (a peripatric scenario). In both scenario's, the sympatric distribution of the Northern Hemisphere clades resulted from secondary contact after divergence in geographic isolation. Alternatively, diversification in the Northern Hemisphere could also have occurred in sympatry.
- Line 616: Distinct clades could be observed within the lineage with the highest amount of sequence data and the widest geographic distribution (reference strain JRI15_10_06, clade 1). AMOVA⁴⁴ was used to test whether this sequence variation was geographically structured between the Northern and the Southern Hemisphere. Since *cox1* showed distinctly more sequence variation than 28S, only the former was used for the AMOVA test. Prior to the analysis, the alignment was reduced in order to include a maximum number of sequences and variable positions, resulting in 68 *cox1* sequences belonging to 17 haplotypes. Following this reduction, all strains from some regions were removed from the dataset, but a sufficient number of strains from both hemispheres were retained to rigorously test for between-hemisphere population differentiation. AMOVA was run using the *amova* function of the R package *ade4*⁹⁶ and using the original, as well as clone-corrected data. The clone-corrected dataset included one representative genotype per population. Significance of the AMOVA tests was assessed using a randomization test (function *randtest*) with 999 permutations. Haplotype networks were visualized using TCS as implemented in PopArt v1.7⁹⁷.

Molecular time-calibration was used to estimate the ages of the different sequence clusters. A subset of the 28S – *cox1* haplotype alignment was obtained, covering four closely related lineages (reference strains JRI15_10_06, (Sterre6)c, MAQ17_160b_03, and JRI15_18b_09) and 48 specimens. *rbcl* was added for those strains for which it was available. The dataset was calibrated in time using BEAST v1.10.4 using an uncorrelated relaxed lognormal clock model, a coalescent constant population size tree prior, and three calibration points with uniform prior distributions (Supplementary Fig. 13). The minimum – maximum boundaries of these calibration points were based on the 95% HSP intervals of the corresponding nodes in the time-calibrated phylogeny of *P. borealis* (Supplementary Fig. 6). Three independent runs of MCMC iterations were implemented for 5 million generations and sampled every 1,000th generation. Convergence and stationarity of the runs was checked in Tracer, after which 10% of the generations were discarded as burnin, all post-burnin trees were combined, and a maximum clade credibility tree with mean node heights was calculated.

(6) Other hypothesis of radiation (e.g. adaptive radiation) need to be acknowledged in the discussion as well as the reason why the authors do not favor this hypothesis (despite the radiating group colonizing a new niche – terrestrial environment (showed by the Supplementary Figure 8) – and this radiation been potentially accompanied by adaption to desiccation and extreme freezing as said in the main text) or both.

Reply. We agree that we should further elaborate on this issue in the manuscript. Evolutionary radiations are often highly complex, and can rarely be explained by one single explanation/driver. This is, for example, nicely pointed out by Simões et al. 2016 (*Trends Ecol. Evol.*) who indicated that many evolutionary radiations seem to be the consequence of multiple factors acting together. For example, many adaptive radiations have been shown to have a geographic factor (see for example the radiation of Darwin’s finches in the Galápagos islands, or the radiation of cichlid fishes in East African lakes).

Generally, adaptive radiations are driven by biotic factors occurring in sympatry and coupled with ecomorphological divergence. In fact, in the literature, ecological speciation is usually associated with sympatric speciation, rather than allopatric speciation. It is generally thought that in an allopatric setting, speciation occurs through independent evolution of geographically isolated populations, prior to the occurrence of potential ecological differentiation between these populations. There are, however, examples of ecological divergence being involved in speciation in an allopatric setting. From this it follows that adaptive radiations predominantly occur in sympatry, as this is the most likely scenario for adaptation to play a dominant role in the speciation process. We believe we have solid evidence that geographic isolation plays an important, if not dominant, role in the diversification history of *P. borealis*. However, this does not mean that ecological speciation in a sympatric setting could not have occurred, and we have made this more clear in the current version of the manuscript.

The reviewer is correct in pointing out that the shift to a new niche (the terrestrial environment) and the associated adaptation to desiccation and freezing could be interpreted as a ‘ecological key’ that has triggered an adaptive radiation. Alternatively, this shift to a new niche could have opened up additional opportunities for allopatric speciation (and thus a geographic radiation), or it is an exaptive trait that became advantageous due to a change in selective regime. Even so, the ‘ecological key’ *per se* is not evidence for an adaptive radiation, as it could have triggered multiple scenario’s. For a more complete overview on these matters, we refer to Simões et al. 2016 (*Trends Ecol. Evol.*).

To better accommodate this complexity and alternative scenarios, we rewrote the last paragraphs of our study to better accommodate for these ideas. Specifically:

- Line 269: Based on the results of our study, two key findings emerge: (i) *P. borealis* shows extraordinarily high species-level diversity, most likely caused by elevated diversification rates, and (ii) diversification is predominantly driven by colonization of novel geographic areas and subsequent evolution in isolation. The existence of hyper-diverse clades is often attributed to evolutionary radiations, the most well-known example of which are adaptive radiations⁴⁶. In

general, adaptive radiations are driven by biotic factors (a key innovation), predominantly occur in sympatry, and are coupled with ecomorphological divergence⁴⁶. There are however other types of evolutionary radiations, and in many cases multiple drivers act together in shaping diversification history⁴⁶. Our analyses on the historical biogeography of *P. borealis* indicate that its diversification has a strong geographical component. This indicates that allopatric speciation plays an important role, suggesting that diversification in this complex is an example of a global-scale geographic radiation. In such a radiation, a clade experiences increased opportunity for diversification due to allopatric speciation resulting from a physical barrier to gene flow⁴⁶. Although allopatric species can be ecologically differentiated, in an allopatric setting ecological speciation resulting from local adaptation usually does not play a (key) role in the speciation process itself. Evidence for this stems from the observation that closely-related allopatric species generally occupy highly similar niches, whereas sympatric species tend to show more niche-divergence^{47,48}. Nevertheless, our results do not exclude the possibility of ecological and/or sympatric speciation as within-region speciation also appears to be important in the *P. borealis* complex. Although it is impossible to determine the relative contributions of sympatric, parapatric and allopatric speciation within the vast continental-scale regions including distant islands defined in our analyses, it at least hints towards a potential opportunity for sympatric speciation to occur. To properly assess the role of sympatric and/or ecological speciation in the *P. borealis* complex, detailed ecological trait information should be obtained for all species, and, if possible, complemented by comparative/population genomic analyses. Nonetheless, it is highly unlikely that sympatric speciation on its own has generated the high levels of diversity of *P. borealis* observed in this study, nor that divergent selection was solely responsible for speciation after the establishment of physical barriers to gene flow. Furthermore, as noted above, it is not unlikely that different drivers are acting together in shaping the extreme species-level diversity observed in *P. borealis*.

(7) L21. check that “have received less study” is proper English. Maybe “have received less interest”?

Reply. This has been adjusted in the text.

(8) L36. Did the authors mean “limited global species richness and wide geographic distributions”? Article 4 claims that microorganisms are ubiquitous.

Reply: Indeed, this has been adjusted, using the word “cosmopolitan”.

(9) L61-63. “Microscopy revealed the presence of *P. borealis* in 29% of the samples in the form of dead (13%) or live (16%) cells, confirming its patchy but widespread geographic distribution.” Could it be present but not seen?

Reply. Although we cannot exclude the possibility that the presence of *P. borealis* has been overlooked, the chance that this has been the case in more than a handful of samples is small. *Pinnularia borealis* is a relatively large diatom with an unmistakable morphology which cannot be confused with other species. For this reason, it is easy to recognize in an environmental sample. Although the abundances of *P. borealis* cells were low in the overall majority of the samples, this was accommodated by careful sample treatment by the first author. All environmental samples were subsampled in multiple wells of 12-well plates. In doing so, care was taken to take material from different parts of the sample, and if the sample was heterogeneous (for example, a mix of soil and moss), multiple subsamples from these different parts were taken. These samples were subsequently screened in a light microscope over a course of multiple weeks - months. In case only dead valves of *P. borealis* were observed, samples were screened over longer time periods (multiple months). Although time-consuming, this approach ensured that the chances of observing living *P. borealis* cells were maximized. We have now further elaborated on our sampling protocol in the Supplementary Methods. Specifically:

→ Upon arrival in the laboratory, small quantities of the natural material (subsamples) were incubated for several weeks to months in WC medium¹, without pH adjustment or vitamin addition, at 4 °C (for polar and temperate regions) or 18 °C (for subtropical regions), 5 – 10 μmol photons m⁻² s⁻¹ and a 12:12h (light:dark) cycle. Although the abundances of *P. borealis* cells were low in the overall majority of the samples, this was accommodated by careful sample treatment. All environmental samples were subsampled in multiple wells of 12-well plates. In doing so, care was taken to take material from different parts of the sample, and if the sample was heterogeneous (for example, a mix of soil and moss), multiple subsamples from these different parts were taken. These samples were subsequently screened repeatedly in a light microscope over a course of several weeks. In case only dead valves of *P. borealis* were observed, samples were screened over longer time periods (up to four to six months). Although time-consuming, this approach ensured that the chances of observing living *P. borealis* cells were maximized.

(10) L77. “Using rarefaction analyses, we estimated the expected diversity of *P. borealis* for the studied regions to be 415 species on average (Fig. 2, Supplementary Fig. 4).” Supplementary 4 (individual-based rarefaction) actually shows a much lower estimation of diversity. Can the authors please correct in the text and explain/interpret the difference between the two estimation results.

Reply. We thank the reviewer for pointing out that this was not clear in the original text. We used two different types of rarefaction analyses. The rarefaction curve in Fig. 2 is a sample-based rarefaction analysis. This analysis investigates how many extra species are to be expected when additional *samples* are investigated. In other words, it investigates the completeness of the sample effort. The rarefaction curve in the supplementary section (Supplementary Fig. 4) is an individual-based rarefaction analysis. This analysis investigates how many extra species are to be expected when additional *strains from the investigated samples* are obtained. In other words, it assesses the completeness of our culture effort. In order to make this more clear, we adjusted the corresponding paragraph in the manuscript accordingly:

→ Line 110: Using sample-based rarefaction analyses, we estimated the expected diversity of *P. borealis* on a global scale to equal 415 species on average (Fig. 2). These additional species are predominantly to be expected when additional samples are investigated, as individual-based rarefaction analyses indicated that within the set of investigated samples, the overall majority of *P. borealis* species present has been found (Supplementary Fig. 4).

(11) L97 and throughout, when reporting diversification rates, please use events per lineage per million year.

Reply. This has been adjusted in the manuscript.

(12) L133-134 “Evolutionary radiations of relatively young and hyper-diverse clades have long fascinated biologists (25), yet have hardly been documented for microorganisms (27,28).” See also Martin et al. (Evolution 2004) for one of the first studies looking at lineage accumulation curves in microbes, and Morlon et al. (Evolution 2012), Louca et al. (Nature Ecol Evol 2018).

Reply. We thank the reviewer for pointing out these interesting studies. Given that our study focuses on protists instead of bacteria, and given that we are limited in the number of references that we can use, we chose only to add one out of three of these references to our manuscript (Morlon et al., as this study is most similar to ours).

(13) Fig 2d: Can the authors use “number of reconstructed lineages” instead of “number of species” or something along these lines (the LTT plot represents, at one given time, the number of lineages that left descendants in the present, not the actual number of lineages/species – all the lineages that did not leave extant descendants are missing). Same remark in the caption, “the exponential accumulation of species” is misleading. The LTT plot can give the impression that there is an exponential accumulation of species, but the true number of species can be very different from the LTT plot (species diversity could follow any type of variation through time, including a curve with increasing diversity followed by decreasing diversity).

Reply. In Fig 2d., ‘number of species’ has been changed into ‘number of reconstructed lineages’, and ‘showing the exponential accumulation of species (constant diversification)’ has been removed from the figure caption.

Reviewer #3 (Remarks to the Author):

I also reviewed the phylogenetics and diversification sections, methods which I am familiar with. Overall it looks to be very professionally done (without having gone through it in extreme detail).

Reply. We thank the reviewer for the valuable feedback, and for giving suggestions to improve our manuscript. Following the suggestions, we made the following changes to the manuscript:

- 1) we discussed potential shortcomings and caveats in our sampling design and the used models;
- 2) we followed all small suggestions given by the reviewer;

A more detailed response to each suggestion can be found below.

There are various detailed controversies about selection of tree models in phylogenetics, and about diversification inferences from trees of all-living species (as is typical for molecular phylogenies). These issues are well-known, however, and the amount we can do about it is pretty limited, due to computational or theoretical limitations. Of all of the issues involved in applying our imperfect models to a very complex reality, probably the most significant is the fact that extinction is often inferred or assumed to be too low, or zero; this issue is best described by Charles Marshall (2017), Nat. Eco. Evol. For the purposes of this paper, I would just make sure there are some caveats described somewhere, and that inferences are thus conditional on the models available.

Reply. Indeed. Following the reviewers suggestion, we now extra information regarding some of the shortcomings of the methods we applied on our dataset, citing, amongst others, Marshall 2017 (*Nat. Eco. Evol.*). Specifically:

- Line 141: The fossil record of *P. borealis* is rudimentary, and its uniform morphology²² precludes direct inference of species turnover rates. Therefore, diversification rate analyses using various models were used to assess diversification rates in the complex based on the time-calibrated phylogeny. Estimating speciation and extinction rates from a phylogeny which only contains extant diversity is a difficult task which is sensitive to undersampling²⁹, and is especially prone to underestimating extinction rates³⁰. As a consequence, the inferred diversification rate estimates are subject to the shortcomings of the chosen models. To, at least partially, alleviate this issue, we chose to adopt a suite of diversification rate models, and assessed congruence amongst the results.

1. "the range evolution model, and the Bayesian Binary Model (BBM) (BAYAREALIKE)"  technically, the BBM model and the BayArea model are different

**** the BBM model just models each area as an independent binary presence-absence character. This was a peculiar invention of the RASP program, where they just threw range data into MrBayes as a series of independent binary characters. One of several weird features of BBM is that you could have an ancestor of "all zeros" (living nowhere).***

**** the BayArea model models range as a series of presences/absences, but uses range-expansion / range-contraction events (just like DEC and DIVA) -- so any new area has been colonized from a different, previously occupied area. This allows e.g. distance to be taken into account in dispersal probability, and disallows an "all zeros" ancestor***

What BBM and BayArea have in common is that they have a very simple cladogenesis model: the entire range is copied from an ancestor to both descendants. This allows e.g. ancestor range ABCDE to produce 2 daughter species with ranges ABCDE. This is biologically implausible in most cases, but it is computationally fast as it avoids the need for any special cladogenesis calculations (BBM and BayArea were both Bayesian

in their original implementation). The BioGeoBEARS "BAYAREALIKE" is an ML implementation of the basic assumptions of BayArea. Short version, this would make more sense: and the BayArea model (BAYAREALIKE).

Reply. We are very grateful for this comment. This issue has been adjusted in the text.

2. Like the diversification & phylogenetics models, lineage extinction is the biggest weakness of the biogeography models. DEC etc. effectively assume a Yule process (pure birth, zero extinction) produced the tree. Matzke (2014) simulations suggest this is not a major issue if extinction is random and speciation > extinction. But, in other situations, it might be. Note also that incomplete sampling is similar to extinction in effect. And, if there are geographic biases in sampling, this could effect inference. The most obvious caveat I can see with the sampling here is relative lack of sampling in terrestrial tropics. I know that we cannot just stop science until sampling is perfect, and that permitting, funds, etc. for sampling globally is highly nontrivial. So I would just ask that the authors have a prominent discussion of the possible caveats due to this sampling issue (mostly, you will usually not infer tropical ancestry if tropical tips are not sampled).

Reply. Many thanks for raising these issues. Indeed, the reviewer is correct that incomplete sampling will have had an influence on our results. We therefore added a few paragraph to the manuscript where we discussed some potential shortcomings in the implications for our results. Specifically:

- Line 240: In light of these results, it has to be noted that our extensive but still incomplete geographic sampling might have affected our results on the historical biogeographical processes shaping the evolutionary history of the *P. borealis* complex, as anagenetic and sympatric processes are likely to be of higher importance in a scenario with wider species distributions. Such wider species distributions may be expected to be uncovered when additional samples/regions would be investigated. However, additional sampling is also likely to uncover additional (rare) species-level diversity with restricted distributions. Furthermore, absence of unsampled species-level diversity in the *P. borealis* phylogeny may have influenced the inference of ancestral ranges, which could in turn have impacted our estimates on historical biogeographical processes. Nevertheless, evidence for a geographic factor in the diversification of *P. borealis* was also uncovered when investigating intraspecific diversity in one of the *P. borealis* species.

(* Also, I was glad to see no mention of the Ree/Sanmartin critique (2018, J. Biogeography) of the DEC/DEC+J comparison; this critique is fatally flawed in numerous ways, including getting the basic likelihood calculations wrong, as anyone can see for themselves if they run the DEC model on the Ree/Sanmartin example trees; both Ree's Lagrange-DEC and Matzke's BioGeoBEARS-DEC produce the same likelihood, but they disagree with the numbers reported in the Ree/Sanmartin paper! For more important (!) problems, like the fact that Ree & Sanmartin totally ignored Matzke (2014)'s published simulation tests of the inference, see Klaus & Matzke, 2019, SysBio for a brief overview.)

Reply. Indeed. However, while we were asked by reviewer 2 to add this particular criticism on the DEC+J model to our manuscript, we did so by citing both sides of the discussion, which will hopefully spur other readers to more critically assess the results of Ree & Sanmartin 2018, and to continue using +J models in their research. Specifically:

- Line 586: It is worth noting that critiques have been raised on the use of the founder-event parameter *j* in DEC models⁹², although others have argued that the latter study shows several fundamental flaws⁹³.

References:

- Alverson, A. J. & Kolnick, L. Intragenomic nucleotide polymorphism among small subunit (18S) rDNA paralogs in the diatom genus *Skeletonema* (Bacillariophyta). *J. Phycol.* **41**, 1248–1257 (2005).
- Godhe, A. *et al.* Quantification of diatom and dinoflagellate biomasses in coastal marine seawater samples by real-time PCR. *Appl. Environ. Microbiol.* **74**, 7174–7182 (2008).
- Hejduková, E. *et al.* Tolerance of pennate diatoms (Bacillariophyceae) to experimental freezing: comparison of polar and temperate strains. *Phycologia* **58**, 382–392 (2019).
- Holt, B. G. *et al.* An update of Wallace’s zoogeographic regions of the world. *Science* **339**, 74–79 (2013).
- Nakov, T., Beaulieu, J. M. & Alverson, A. J. Accelerated diversification is related to life history and locomotion in a hyperdiverse lineage of microbial eukaryotes (Diatoms, Bacillariophyta). *New Phytol.* **219**, 462–473 (2018).
- Parks M. B., Wickett N. J. & Alverson A. J. Signal, Uncertainty, and Conflict in Phylogenomic Data for a Diverse Lineage of Microbial Eukaryotes (Diatoms, Bacillariophyta). *Mol. Biol. Evol.* **35**, 80-93 (2018).
- Simões, M. *et al.* The evolving theory of evolutionary radiations. *Trends Ecol. Evol.* **31**, 27–34 (2016).
- Stock, W. *et al.* Expanding the toolbox for cryopreservation of marine and freshwater diatoms. *Sci. Rep.* **8**, 4279 (2018).
- Theriot E. C. *et al.* A preliminary multigene phylogeny of the diatoms (Bacillariophyta): challenges for future research. *Plant Eco. Evol.* **143**, 278-296 (2010).

Reviewers' Comments:

Reviewer #1:

Remarks to the Author:

This second version of the manuscript has been improved considerably since the first, and authors have, in my opinion, addressed properly all queries. For me, the manuscript can be published as it is.

Reviewer #2:

Remarks to the Author:

The authors did a great job at addressing comments from us and other reviewers. We are satisfied with their answers, new analyses, and edits in the text. We only have two additional comments, detailed below.

1) Throughout the paper, it is implicitly assumed that *P.borealis* species are mostly terrestrial. Is there good support for this, and if yes, can the authors mention it explicitly and provide references? If *P.borealis* species are found in freshwater there could have been several independent transitions from/to freshwater and terrestrial environments that would affect the hypothesis emitted in the present study (long distance dispersal and colonization of *P.borealis* species as a potential cause of allopatric speciation) and this should be discussed.

2) There is a new implementation of ClADS (including ClADS2) with data augmentation that is much faster and would converge very rapidly on the author's data

https://github.com/hmorlon/PANDA/tree/ClADS_DataAugmentation/ClADS_DA

We totally understand if the authors don't want to rerun their analyses with ClADS2 with this new implementation, but they could mention that it exists in their Methods section.

Minor: Could the authors use something else than "clade" at the intra-specific level (Line 260 and Line 398), "population" for example?

This response letter contains the changes made upon the second revision of the manuscript entitled “Global radiation in rare biosphere soil diatom”. Below, the requests of the reviewers are addressed in detail.

Response to reviewers:

Reviewer #1:

This second version of the manuscript has been improved considerably since the first, and authors have, in my opinion, addressed properly all queries. For me, the manuscript can be published as it is.

Reviewer #2:

The authors did a great job at addressing comments from us and other reviewers. We are satisfied with their answers, new analyses, and edits in the text. We only have two additional comments, detailed below.

1) Throughout the paper, it is implicitly assumed that *P. borealis* species are mostly terrestrial. Is there good support for this, and if yes, can the authors mention it explicitly and provide references? If *P. borealis* species are found in freshwater there could have been several independent transitions from/to freshwater and terrestrial environments that would affect the hypothesis emitted in the present study (long distance dispersal and colonization of *P. borealis* species as a potential cause of allopatric speciation) and this should be discussed.

Reply. Yes, the *P. borealis* complex is typically regarded as a soil diatom complex. Whereas it is widely cited in the literature focusing on soil algae (e.g. Lund 1945, Johansen et al. 1982, Van de Vijver & Beyens 1997, van Kerckvoorde et al. 2000, Van de Vijver et al. 2002, Falasco et al. 2014, Barragán et al. 2017, Sommer et al. 2020), it is absent from the overall majority of studies on freshwater floras. This is also true in our own experience: when examining extensively the soils and lakes within the same area, Pinseel et al. (2017a) never observed *P. borealis* cells in ponds and lakes in the Petuniabukta region (Svalbard), whereas we had several terrestrial soil and moss samples from the same region that harbored populations of *P. borealis*. That being said, *P. borealis* is occasionally found in morphology-based surveys of freshwater diatom flora's (e.g. Sabbe et al. 2003, Van de Vijver & Zidarova 2011, Kopalová & Van de Vijver 2013), but in these cases it never constitutes a dominant part of the diatom flora. Furthermore, such morphology-based surveys fail to determine whether the *P. borealis* cells are actually alive in these freshwater environments because diatom cells are oxidized prior to data analysis, preventing distinction between living and dead material. It could well be that many of these observations represent *P. borealis* cells that were washed in from surrounding soils and moss vegetation, rather than actual active populations growing *in situ*.

However, we also know that *P. borealis* can sometimes be found alive in the shallow littoral benthic zone of freshwater bodies. This occurrence in shallow freshwater environments is mostly apparent in, but not restricted to, the Continental and Maritime Antarctic region (own observations). This is likely because the Antarctic diatom flora is highly impoverished and predominantly consists of typical terrestrial diatom genera, whereas obligate freshwater genera are represented by very few and mostly endemic species (Verleyen et al. in prep.). We hypothesize that the absence of a typical diverse freshwater flora in Antarctica likely creates niche availability due to reduced interspecific competition, and has allowed *P. borealis* and other terrestrial diatoms to secondarily colonize shallow aquatic environments. To account for a potential freshwater occurrence of *P. borealis*

species, we have included several aquatic samples in our analysis (see Supplementary Data 1 and Supplementary Data 3). When mapping habitat type on the phylogeny, it became clear that all species that occurred in aquatic environments were also found in terrestrial environments, with three exceptions (Supplementary Fig. 9). One of these exceptions is the species *P. catenaborealis* which was discussed by Pinseel et al. (2017b) and is restricted to the Maritime Antarctic. This species forms chains by means of spines, and might actually be adapted to living in a freshwater environment as it has not been recovered from soil samples. However, *P. catenaborealis* is also highly tolerant to freezing events as it has been observed to survive -180°C during freezing experiments (Hejduková et al. 2019), suggesting it retained its ancestral adaptation to a terrestrial lifestyle. In addition, the observation that the majority of the *P. borealis* species found in shallow aquatic environments are also found in terrestrial environments suggests that these species are not pre-adapted to an aquatic environment *per se*, but happen to occur in aquatic environments whenever the biotic and abiotic conditions allow for it. Furthermore, it is not always possible to make a clear distinction between an aquatic and terrestrial habitat. For example, ponds can be ephemeral and littoral zones of lakes can fall dry when the water volume is decreasing, thus more closely resembling a terrestrial environment. In this study, aquatic environments were defined as 'being submerged at the time of sampling', but we could not account for potential fluctuations in water level throughout the year. Altogether, this suggests that although some transitions to a truly aquatic lifestyle might have occurred (cf. *P. catenaborealis*). Therefore, we believe that the occasional occurrence of *P. borealis* in shallow aquatic environments will not have affected our general conclusions. Nevertheless, we agree with the reviewer that we should be more clear about the habitat type of *P. borealis* species, and we adapted the text accordingly:

- ➔ Line 176: Species belonging to the diatom genus *Pinnularia* are confined to aquatic habitats, such as the shallow littoral zones of freshwater ponds and lakes, as well as semiterrestrial habitats such as wet soils and mosses¹⁶. In contrast, extant *P. borealis* species are found in drier, truly terrestrial, habitats¹⁶, although some rare exceptions exist²⁰. Our ancestral habitat reconstruction showed that this transition from an aquatic to a terrestrial lifestyle likely happened in the ancestor of the *P. borealis* clade (Fig. 2a, Supplementary Fig. 9).
- ➔ Line 188: In this study, four *P. borealis* species were found to inhabit submerged shallow-water habitats in addition to terrestrial environments, and three were only detected in such submerged habitats, including the recently described species *P. catenaborealis*²⁰ (Supplementary Fig. 9). Such occurrences in aquatic environments were most pronounced in the Antarctic region, and could reflect niche availability due to reduced interspecific competition resulting from the generally low species-diversity of freshwater diatoms in this region (Verleyen & Van de Vijver, own observations). In addition, shallow ponds and the littoral zones of lakes can be highly ephemeral and susceptible to desiccation, thus representing a more terrestrial environment. Interestingly, the aquatic species *P. catenaborealis* was found to exhibit high tolerance to freezing stress³³, suggesting that although some truly aquatic *P. borealis* species might exist, they likely retained the ancestral adaptations to a terrestrial lifestyle. Indeed, our analyses suggest that occurrences of *P. borealis* in shallow aquatic environments represent recent secondary colonizations from a terrestrial ancestor (Supplementary Fig. 9). Such colonizations might have been triggered whenever the (a)biotic conditions, such as reduced interspecific competition, allowed for it.
- ➔ Line 311: In addition, evidence for local adaptation comes from the observation that (i) different *P. borealis* species show niche-divergence regarding optimal growth temperatures and maximum temperature for growth¹⁸, and (ii) *P. catenaborealis*, a chain-forming *P. borealis* species that is restricted to Maritime Antarctica, is likely adapted to a truly aquatic lifestyle²⁰.

In addition, the reviewer asked to more clearly state that the *P. borealis* complex is a terrestrial diatom species complex, and provide references for this statement. Since we are restricted on the number of references that can be used in the manuscript, we chose to refer only add one new reference to the manuscript. This reference contains more detailed information as well as other references, so we believe this is sufficient for the reader to retrieve the necessary information from the literature:

- ➔ Line 65: Species belonging to the *P. borealis* complex predominantly live in terrestrial habitats such as moist to dry soils and mosses¹⁶.

Also, it became clear to us that we did not clearly specify in the original manuscript how the habitat type of the species was determined. Therefore we added the following sentence to the header of Supplementary Fig. 9, which also includes a reference towards *P. catenaborealis* which was referred to in the main text:

→ The habitat type identification of each species was based on the habitat type of all individual strains incorporated in the phylogeny. The presumed aquatic species *P. catenaborealis* is referred to as reference strain EVA_P3.

We also updated Supplementary Fig. 9. The new figure includes the names of the strains and has a larger size, which will make it easier for the reader to link the results with the other figures in the manuscript.

2) There is a new implementation of ClaDS (including ClaDS2) with data augmentation that is much faster and would converge very rapidly on the author's data https://github.com/hmorlon/PANDA/tree/ClaDS_DataAugmentation/ClaDS_DA. We totally understand if the authors don't want to rerun their analyses with ClaDS2 with this new implementation, but they could mention that it exists in their Methods section.

Reply. We thank the reviewer for pointing this out. As suggested, we added a sentence to the methods section, mentioning this new version of ClaDS.

→ Line 582. Finally, it is worth nothing that at the time of publication of this manuscript, a new version of ClaDS had become available that allows for data augmentation, and promises to substantially speed-up analyses of large datasets with missing data under the ClaDS2 model.

Minor: Could the authors use something else than “clade” at the intra-specific level (Line 260 and Line 398), “population” for example?

Reply. We have now changed the use of ‘clade’ into ‘metapopulation’ when referring to the intra-specific level. We chose the term ‘metapopulation’, as we think it properly identifies the reality of the different clusters on the intra-specific level.

References:

Barragán C., Wetzel C.E. & Ector L. A standard method for the routine sampling of terrestrial diatom communities for soil quality assessment. *J. Appl. Phycol.* **30**, 1095–1113 (2017).

Falasco E., Ector L., Isaia M., Wetzel C.E., Hoffmann L. & Bona F. Diatom flora in subterranean ecosystems: a review. *Int. J. Spel.* **43**, 231–251 (2014).

Hejduková E. et al. Tolerance of pennate diatoms (Bacillariophyceae) to experimental freezing: comparison of polar and temperate strains. *Phycologia* **58**, 382–392 (2019).

Johansen J.R., Javakul A. & Rushforth S.R. Effects of burning on the algal communities of a high desert soil near Wallsburg, Utah. *J. Range Man.* **35**, 598–600 (1982).

Kopalová, K., Van de Vijver B. Structure and ecology of freshwater benthic diatom communities from Byers Peninsula, Livingston Island, South Shetland Islands. *Ant. Science* **25**, 239–253 (2013).

Lund J.W.G. Observations on soil algae. I. The ecology, size and taxonomy of British soil diatoms. *The New Phytol.* **44**, 196–219 (1945).

Pinseel, E. et al. Diversity, ecology and community structure of the freshwater littoral diatom flora from Petuniabukta (Spitsbergen). *Polar Biology* **40**, 533–551 (2017a).

Pinseel, E. et al. *Pinnularia catenaborealis* sp. nov. (Bacillariophyceae), a unique chain-forming diatom species from James Ross Island and Vega Island (Maritime Antarctica). *Phycologia* **56**, 94–107 (2017b).

Sabbe K., Verleyen E., Hodgson D.A., Vanhoutte K. & Vyverman W. Benthic diatom flora of freshwater and saline lakes in the Larsemann Hills and Rauer Islands, East Antarctica. *Ant. Science* **15**, 227–248 (2003).

Sommer, V. et al. Halophilic algal communities in biological soil crusts isolated from potash tailings pile areas. *Frontiers in Ecology and Evolution*. doi: 10.3389/fevo.2020.00046 (2020) (*in this study, P. borealis was misidentified as Surella sp.*).

Van de Vijver B. & Beyens L. The epiphytic diatom flora of mosses from Stromness Bay area, South Georgia. *Pol. Biol.* **17**, 492–501 (1997).

Van de Vijver B. & Zidarova R. Five new taxa in the genus *Pinnularia* section *Distantes* (Bacillariophyta) from Livingston Island (South Shetland Islands). *Phytotaxa* **24**, 39–50 (2011).

Van de Vijver B., Ledeganck P. & Beyens L. Soil diatom communities from Ile de la Possession (Crozet, sub-Antarctica). *Pol. Biol.* **25**, 721–729 (2002).

van Kerckvoorde A., Trappeniers K., Nijs I., Beyens L. Terrestrial soil diatom assemblages from different vegetation types in Zackenberg (Northeast Greenland). *Pol. Biol.* **23**: 392–400 (2000).